# Prevalence and incidence of diabetic peripheral neuropathy in Latin America and the Caribbean: A systematic review and meta-analysis

Marlon Yovera-Aldana[1]*, Victor Velásquez-Rimachi[1,2,3], Andrely Huerta-Rosario[1,2,4], M. D. More-Yupanqui[2,5], Mariela Osores-Flores[2,3], Ricardo Espinoza[6], Fradis Gil-Olivares[2,7], César Quispe-Nolazco[2], Flor Quea-Vélez[2,8], Christian Morán-Mariños[2,9], Isabel Pinedo-Torres[1,2,10], Carlos Alva-Diaz[1,2,11], Kevin Pacheco-Barrios[12,13]

**1** Grupo de Investigación Neurociencia, Efectividad y Salud Pública, Universidad Científica del Sur, Lima, Perú, **2** Red de Eficacia Clínica y Sanitaria, REDECS, Lima, Perú, **3** Facultad de Medicina, Universidad Nacional Mayor de San Marcos, Lima, Perú, **4** Facultad de Medicina Hipólito Unanue, Universidad Nacional Federico Villarreal, Lima, Perú, **5** Servicio de Patología, Departamento de Ayuda Diagnóstico, Hospital Daniel Alcides Carrión, Callao, Perú, **6** Escuela de Medicina, Universidad Peruana de Ciencias Aplicadas, Lima, Perú, **7** Unidad de Guías de Práctica Clínica, AUNA, Lima, Perú, **8** Facultad de Ciencias de la Salud, Universidad Privada San Juan Bautista, Lima, Perú, **9** Unidad de Investigación en Bibliometría, Vicerrectorado de Investigación, Universidad San Ignacio de Loyola, Lima, Perú, **10** Servicio de Endocrinología, Departamento de Medicina y Oficina de Apoyo a la Docencia e Investigación (OADI), Hospital Daniel Alcides Carrín, Callao, Perú, **11** Servicio de Neurología, Departamento de Medicina y Oficina de Apoyo a la Docencia e Investigación (OADI), Hospital Daniel Alcides Carrión, Callao, Peru, **12** Unidad de Investigación para la Generación y Síntesis de Evidencias en Salud, Universidad San Ignacio de Loyola, Lima, Perú, **13** Neuromodulation Center and Center for Clinical Research Learning, Spaulding Rehabilitation Hospital and Massachusetts General Hospital, Harvard Medical School, Boston, Massachusetts, United States of America

* myovera@cientifica.edu.pe

**Data Availability Statement:** All relevant data are within the manuscript and its Supporting Information files.

## Abstract

### Aims

The objective of this systematic review and meta-analysis is to estimate the prevalence and incidence of diabetic peripheral neuropathy (DPN) in Latin America and the Caribbean (LAC).

### Materials and methods

We searched MEDLINE, SCOPUS, Web of Science, EMBASE and LILACS databases of published observational studies in LAC up to December 2020. Meta-analyses of proportions were performed using random-effects models using Stata Program 15.1. Heterogeneity was evaluated through sensitivity, subgroup, and meta-regression analyses. Evidence certainty was performed with the GRADE approach.

### Results

Twenty-nine studies from eight countries were included. The estimated prevalence of DPN was 46.5% (95%CI: 38.0–55.0) with a significant heterogeneity ($I^2$ = 98.2%; p<0.01). Only two studies reported incidence, and the pooled effect size was 13.7% (95%CI: 10.6–17.2). We found an increasing trend of cumulative DPN prevalence over time. The main sources

**Funding:** The author(s) received no specific funding for this work.

**Competing interests:** The authors have declared that no competing interests exist.

of heterogeneity associated with higher prevalence were diagnosis criteria, higher A1c (%), and inadequate sample size. We judge the included evidence as very low certainty.

## Conclusion

The overall prevalence of DPN is high in LAC with significant heterogeneity between and within countries that could be explained by population type and methodological aspects. Significant gaps (e.g., under-representation of most countries, lack of incidence studies, and heterogenous case definition) were identified. Standardized and population-based studies of DPN in LAC are needed.

## Introduction

Diabetes mellitus (DM) is an important global health issue. Around 425 million people worldwide are suffering from this disease, and this number is expected to rise to 628 million people by 2045 [1]. Diabetic peripheral neuropathy (DPN) is the most prevalent complication of diabetes mellitus [2]. The prevalence of DPN ranges from 21.3 to 34.5% in type 2 DM (T2DM) [3–6] and between seven to 34.2% in type 1 DM (T1DM) [7–10]. Of these, up to 45% of patients with type 2 DM and 54% with type 1 DM could be asymptomatic [1, 11].

DPN refers to disorders affecting the peripheral nervous system [12]. The most common presentation is distal symmetric polyneuropathy, typically associated with numbness, tingling [13], pain [14], or weakness [15] that begins in the feet and spread proximally in a stocking distribution [1, 16]. DPN is a leading cause of worldwide disability [15], and it affects the quality of life due to chronic pain, high risk of falls [17], foot ulceration [18], and limb amputation [19]. Furthermore, DPN symptoms often lead to sleep disorders, anxiety, and depression [20, 21]. The poor glycemic control causing hyperglycemia and microangiopathy is the common underlying pathophysiology. However, other factors are involved in the neuropathy progression, such as modifiable cardiovascular risk factors, including dyslipidemia, smoking, and hypertension [22]; consequently, public health strategies could be implemented to reduce the disease frequency. Despite DPN's importance, effective screening methods are lacking, which results in a diagnostic delay of DPN [23, 24], hence producing heterogeneous epidemiological estimates between regions.

A systematic assessment of the DPN distribution and its epidemiological features is crucial to develop public health interventions to control the disease; however, few studies have performed a systematic literature review on the topic [12, 25–27]. A small subset used meta-analytic methods to assess the heterogeneity of the DPN epidemiology [25–27], and none of them use evidence certainty assessment to critically evaluate the body of evidence. To our knowledge, no previous studies have addressed this question in Latin America and the Caribbean (LAC), a region with mainly developing countries where access to healthcare could influence the adequate glycemic control and cardiovascular risk factors of people with diabetes mellitus, and thus affecting the DPN epidemiology. The present study aims to estimate the prevalence of DPN in LAC, assess the evidence certainty of the estimates, and use meta-analytic techniques to explore their heterogeneity sources. This is a necessary starting point in discussing the prevention and diagnosis gaps of DPN in this region.

## Materials and methods

We guided this protocol by the guidelines of the "Preferred Reporting Items for Systematic Reviews and Meta-Analyses" (PRISMA) and some recommendations from the Cochrane

Handbook for Systematic Reviews of Interventions [28, 29] (**S1 Checklist**). The study protocol has been registered at PROSPERO, number CRD42019148273 [30].

## Eligibility criteria

We defined DPN as symmetrical sensorimotor polyneuropathy caused by metabolic and microvascular alterations, such as chronic hyperglycemia exposure and cardiovascular risk factors in patients with DM.

Articles were included if they met the following criteria: (a) studies reported their outcome variable as prevalence or incidence of DPN in the LAC population. We also consider Diabetes complications that included a description of DPN. We included both T1DM and T2 DM, as well as mixed population studies, due to the reported difficulties for a correct diagnostic differentiation in the region (unavailability of T1DM-specific antibodies tests, T2DM predominance, and new DM subtypes/phenotypes) (ref), and potential risk of diagnosis overlap [31, 32] (b) peer-reviewed journal articles (c) any language (d) cross-sectional or cohort studies (e) diagnosis with at least two physical tests (f) any time of publication. We excluded: (a) another kind of neuropathy such as carpal tunnel syndrome, cardiac autonomic neuropathy, cranial neuropathy (b) subsequent stages of diabetes mellitus such as diabetic foot or need for hospitalization; (c) unclear peripheric neuropathic criteria (d) cases report, case series, case-control study.

## Literature search and study selection

We carried out a systematic search in five databases: Medline, Scopus, Web of Science, Embase, and Scielo. As recommended by Cochrane collaboration, we included additional relevant articles from other sources via a hand search of grey literature and other related articles due to the low rate of database indexation of regional journals [33].

We performed our database searches on October 15th, 2019, and updated this search on December 14th, 2020, to find additional eligible studies to be included. Our search strategy included Medical Subject Title (MeSH) terms and free-text terms such as "diabetic neuropathies," "prevalence," "incidence," and "Latin-America." Boolean operators like "AND" and "OR" were used to combine search terms. We include observational studies without restrictions regarding language. The complete search strategy is available in the **S1 Table**.

According to the inclusion criteria, two independent authors (MMY and MOF) selected articles by titles and abstracts to identify potentially relevant articles. One of the authors (MMY) handled the duplicates. Lastly, the same authors accessed the full-text articles and evaluated their eligibility for inclusion. A third author (MYA) addressed the missing data and resolved discrepancies by discussion and consensus. The complete list of excluded articles at this full-text stage is available in **S2 Table**.

**Data extraction.** Two independent researchers (MOF and MYA) extracted the following information from each of the included studies into a Microsoft Excel sheet: author, year of publication, country, design, center, type of population, sex, range age, diabetes time, setting, diagnostic type, sample size, and reported prevalence. A third author (AHR) checked that the data was correct. Only, we included the most recent or complete publication when we identified studies with the same population. In case any study showed more than one prevalence by different methods, we choose the prevalence with the highest performance.

Considering the heterogenous DPN definition, we extracted the case definitions from each included study. Then, we established the DPN diagnosis according to the Toronto Diabetic Neuropathy Expert group's definitions: (a) confirmed DPN (abnormal nerve conduction and a symptom or sign of neuropathy), (b) probable DPN (a combination of symptoms and signs

of neuropathy), (c) possible DPN (any symptoms or signs) and (d) subclinical DPN (no signs or symptoms of neuropathy with abnormal nerve conduction test or a validated measure of small fiber neuropathy [34]).

**Risk of bias assessment.** Two researchers (MYA and MOF) evaluated the methodological quality of the prevalence studies according to the questionnaire developed by Loney et al. [35], a third author (AHR) settled in case of doubt. Eight criteria were evaluated and scored with one point if it existed: (a) random sample or whole population; (b) an unbiased sampling frame (i.e., census data); (c) adequate sample size > 323 subjects, considering the prevalence of DPN of 30% according to Sun et al. [27], 5% alpha, and 80% of power; (d) measures were performed with the diagnostic standard for diabetic neuropathy (clinical and electromyography assessment); (e) outcomes were measured by the unbiased assessor; (f) reasonable response rate (>70%) and refusers described; (g) confidence intervals and subgroup analysis were described; and (h) participants' characteristics were described. We calculated the total score for each study (score range 0–8). We considered a quality score of 0–2 as very low, 3–4 as low, 5–6 as moderate, and a score of 7–8 as high. Cohort studies were evaluated based on the New-castle-Ottawa Scale (NOS). This tool consists of 8 items grouped into three main components: selection, comparability, and outcome [36].

## Statistical analyses

We calculated the pooled DPN prevalence with the corresponding 95% confidence intervals (CI), expressed as the percentage of diabetic patients with DPN. We used the Freeman-Tukey Double Arcsine transformation to stabilize the proportion variances before performing the pooled analysis [37]. According to the high expected between-study heterogeneity, a prespecified random-effects model performed the meta-analysis due to the DerSimonian, and Laird method [38], and 95% CI calculation were based on the exact method [39]. We assessed the presence of between-study heterogeneity using Cochran's Q chi-square statistics and quantified using the $I^2$ statistical test. [40, 41]. The $I^2$ statistic was calculated only in a subgroup of four or more included studies, as recommended by previous studies [42–44], due to the underestimation of heterogeneity in small meta-analyses. For meta-analysis with less than four included studies, we assessed between studies heterogeneity visually, looking at the confidence intervals overlap. For studies with four or more included studies, an $I^2$ value> 75% was interpreted as high heterogeneity [45], to assess the trend of the pooled DPN estimate across time, we performed and plotted a random-effects cumulative meta-analysis based on the publication year of each included study.

To evaluate the sources of heterogeneity among the primary studies, we carried out subgroups analysis according to country, population type, diabetes type, age group, time of diabetes, and the Toronto DPN Study Group criteria. We also conducted a sensitivity analysis to assess the estimates' robustness by evaluating the influence of any individual study and important methodological variables: type of design, sampling, type of recollection, blind evaluation, size sample, and quality score.

Additionally, we conducted univariate random-effects meta-regression to test study level moderators of the DPN prevalence [46]. We based our analysis on the Thompson and Higgins recommendations [47]. Each moderator was tested on a minimum of eight included studies in the meta-analysis [47]. We used an entry criterion of p = 0.2 for independent variables. We set a high p-value to reduce the risk of omitting potential confounders variables. We used a step-down method to build the multivariate model. To select the best model, we assessed the residual percentage of variation due to heterogeneity ($I^2$, $Tau^2$) and the proportion of between-study variance explained (adjusted $R^2$), in addition to the significant criterion of p<0.05 per

each moderator. We used these covariates as follows: age, time of diabetes, A1C, year of publication, sample size, type of DPN according to the Toronto criteria, and quality score. We performed a Monte Carlo permutation test to account for high false-positive rates associated with meta-regression models (i.e., repeated random sampling) using 10,000 random permutations.

Furthermore, we assessed publication bias by visual inspection of a funnel plot of the standard error. We performed the Egger's regression test and the trim and fill method for publication bias, calculating the linear estimator [48]. We performed all statistical analyses using STATA 15.

**Geographical assessment.**   We showed the geographical representation of the prevalence estimates and each country's research production using a map downloaded from https://yourfreetemplates.com/ with Attribution-No-Derivatives 4.0 International (CC BY-ND 4.0) Creative Commons' license.

**Evidence certainty assessment.**   We assessed the certainty of our DPN prevalence and incidence estimates in LAC using the grading of recommendation, assessment, development, and evaluation (GRADE) approach [49]. We based this critical appraisal on five domains: study limitations (risk of bias of the studies included), imprecision (sample sizes and CI), indirectness (generalizability), inconsistency (heterogeneity), and publication bias, as stated in the GRADE handbook [50]. We adapted the assessment to prevalence estimates. The evidence's certainty was characterized as high, moderate, low, or very low [49]. The results were reported as a summary of findings table (SoF), adapted manually from the GRADE online tool (http://gradepro.org).

## Results

### Search results

We identified 2180 titles, 1903 during the initial search, 13 titles for expert recommendation from manual search, and 164 from the second search with publication purposes. A total of 70 full-text studies were read and assessed. After applying the pre-defined criteria, we excluded forty-one studies (**S2 Table**). Finally, we included twenty-nine studies in the final analysis [51–79]. **Fig 1** shows a flow chart to illustrate the process of article selection and inclusion in the study.

**Study characteristics.**   We included 28 studies with 8139 subjects for DPN prevalence (**Table 1**), and included two studies for DPN incidence. Cardoso et al. study was used in both frequencies (**Table 2**).

Brazil had the most scientific production with fourteen studies [53, 55–58, 60, 61, 64–66, 71, 72], Mexico with seven [52, 54, 67, 68, 70, 73, 74], Peru with four [59, 69, 74, 77], and Cuba [51], Uruguay [63], and Ecuador with one study [62] (**Fig 2**). Regarding DM type, 22 studies included type 2 DM [52–57, 59–62, 66, 68–78], three type 1 DM [58, 65, 67] and three in both types of DM [51, 63, 64], presenting combined prevalence. In twenty-one studies, patients were recruited from reference hospitals [51, 54–59, 61, 63–67, 69, 70, 72–74], four from primary care [52, 53, 60, 68], and three from the general population [62, 71, 75]. According to the onset of DM, one study was made with newly detected DM (debut) [61], nine with DM patients with more than 5 years of illness [51, 56, 65, 68–70, 72, 77, 78], six with more than 10 years of disease [53, 57, 64, 66, 73, 74], and eleven did not specify any time [52, 54, 55, 58–60, 62, 63, 71, 75, 76].

There were several methods for DPN diagnosis, we classified in four groups: a) >2 clinical signs, including nine studies [48, 49, 54, 56, 58, 64, 71, 72, 75]; b) >2 clinical signs and symptoms, including 14 studies [50, 52, 53, 57, 59–61, 63, 65, 66, 68–70]; c) >2 clinical signs + nerve conduction test (NCT), including one study [55]; and d) only NCT or sudomotor disfunction test, including four studies [51, 62, 67, 74]. According to Toronto Study Group, one study

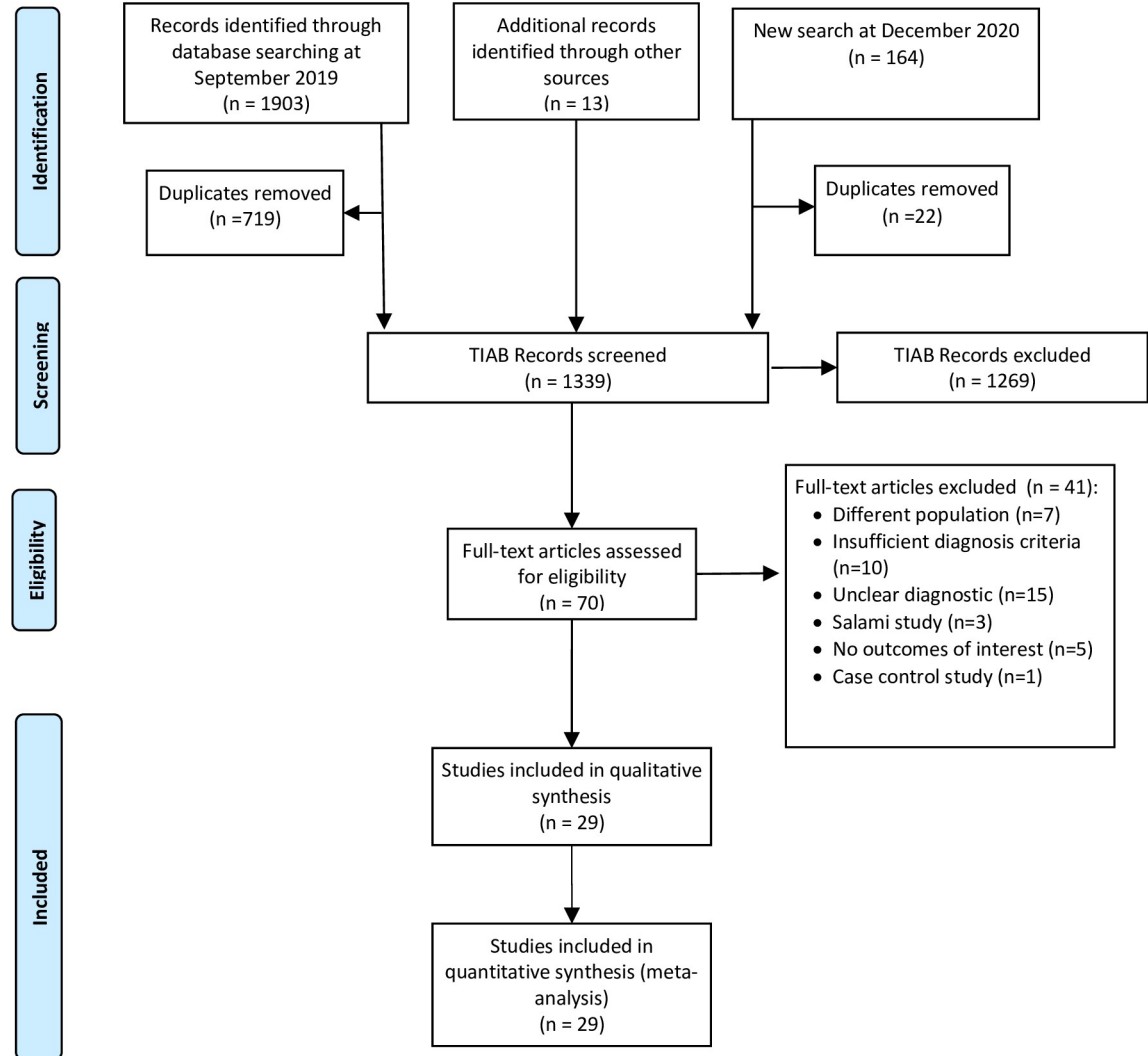

**Fig 1. Flow chart for the selection of included studies.**

included confirmed DPN cases [55], 20 included probable DPN cases [48–50, 52–54, 57, 59–61, 63–66, 68–73, 75], three studies included possible DPN [56, 58, 72], and four included sub-clinical DPN [51, 62, 67, 74].

## Risk of bias of included studies

The quality score range of the prevalence studies of DPN was from two to eight. There was one study of very low quality [50], 14 clinical studies were of low quality [48, 51, 54, 55, 60–62, 64, 66–69, 71, 72], eight studies were of moderate quality [49, 52, 53, 57, 74, 75], and seven studies were of high quality [56, 58, 59, 63, 65, 70, 73]. The two studies included for incidence obtained a score of six according to the Newcastle-Ottawa scale [53, 76] (**S3 and S4 Tables**).

## Pooled estimates and cumulative meta-analysis

We found a pooled DPN prevalence in LAC of 46.5% (95% CI: 38.0 to 55.0) in 28 studies. Significant heterogeneity was identified among studies ($I^2$ = 98.24%, p < 0.001) (**Fig 3**). All these

**Table 1. Characteristics of 28 included studies of diabetic peripheral neuropathy prevalence in Latin America and the Caribbean.**

| | Author (year) | Country | Design | Population, age (mean-years), male (%), diabetes time (median-years), A1c % (mean) | DM type | Primary outcome | Diagnostic criteria according to the study | Grouping criteria | Toronto Diabetic Neuropathy Expert Group Criteria | Sample size (N) | Diabetic neuropathy cases (n) | Prevalence (%) | Quality assessment (total score) |
|---|---|---|---|---|---|---|---|---|---|---|---|---|---|
| 1 | Alvarez (2015) [51] | Cuba | Cross-sectional | Population: reference center; Age: 51; Male n(%): 89 (39.7%); Diabetes time: 9.88; A1c%: NR | T1DM/ T2DM | Diabetes complications | ≥ 2/9 physical exam | Physical exam ≥ two signs | Probable | 224 | 135 | 63.70% | 4 |
| 2 | Arellano (2018) [52] | Mexico | Cross-sectional | Population: primary care; Age: 59; Male n(%): 63 (59.4); Diabetes time: NR; A1c%: NR | T2DM | Peripheral neuropathy | MNSI physical exam ≥ 2/ 10 | Physical exam ≥ two signs | Probable | 106 | 86 | 81.1% | 5 |
| 3 | Barrile (2013) [53] | Brazil | Cross-sectional | Population: primary care; Age: ; Male n(%): 26 (38.2); Diabetes time: 10.7 years; A1c%: 7.7 | T2DM | Peripheral neuropathy | TCNS + monofilament | Physical exam ≥ two signs + symptoms | Probable | 68 | 39 | 57.3% | 2 |
| 4 | Carbajal Ramirez (2019) [54] | Mexico | Cross-sectional | Population: reference center; Age: 59.8; Male n(%): 73 (33.0); Diabetes time: NR; A1c%: NR | T2DM | Peripheral neuropathy | Sudomotor dysfunction | Autonomic sudomotor dysfunction | Confirmed/Sub clinic | 221 | 134 | 60.6% | 4 |
| 5 | Cardoso (2018) [55] | Brazil | Cohort | Population: reference center; Age: 60; Male n(%): 262 (39.2); Diabetes time: NR; A1c%: 7.7 | T2DM | Diabetes complications | NSS and NDS: moderate symptoms with/without signs or mild symptoms + moderate signs | Physical exam ≥ two signs + symptoms | Probable | 668 | 196 | 29.2% | 5 |
| 6 | Cardoso (2008) [56] | Brazil | Cohort | Population: reference center; Age: 60.5; Male n(%): 250 (53.1); Diabetes time: 9.3; A1c%: NR | T2DM | Diabetes complications | ≥ 2/4: symptoms, monofilament, tuning-fork test, altered reflexes | Physical exam ≥ two signs + symptoms | Probable | 471 | 68 | 14.4% | 5 |

*(Continued)*

**Table 1.** (Continued)

| | Author (year) | Country | Design | Population, age (mean-years), male (%), diabetes time (median-years), A1c % (mean) | DM type | Primary outcome | Diagnostic criteria according to the study | Grouping criteria | Toronto Diabetic Neuropathy Expert Group Criteria | Sample size (N) | Diabetic neuropathy cases (n) | Prevalence (%) | Quality assessment (total score) |
|---|---|---|---|---|---|---|---|---|---|---|---|---|---|
| 7 | Cardoso (2020) [57] | Brazil | Cross-sectional | Population: Reference center; Age: 59.6; Male n(%): 30 (35.3); Daibetes time: 14.5; A1c%: | T2DM | Diabetes complications | LOPS: 10 g monofilament + least 1 altered (128 Hz tuning fork, pinprick sensa-tion and/or an ankle reflex) | Physical exam ≥ two signs | Probable | 85 | 50 | 58.8% | 4 |
| 8 | Coutinho (2002) [58] | Brazil | Cross-sectional | Population: reference center; Age: 13.0; Male n(%): 18 (64.3); Diabetes time: NR; A1c%: NR | T1DM | Peripheral neuropathy | ≥ 2/4: symptoms, signs, bio-thesiometer, nerve conduction test | Physical exam ≥ two signs + symptoms + Nerve conduction test | Confirmed | 28 | 8 | 28.0% | 4 |
| 9 | Damas (2017) [59] | Peru | Cross-sectional | Population: reference center; Age: 60.3; Male n(%): 96 (25.1); Diabetes time: NR; A1c%: NR | T2DM | Diabetes complications | Monofilament with or without tuning-fork test | Physical exam ≥ two signs | Possible | 382 | 131 | 35.50% | 7 |
| 10 | De Matos (2020) [60] | Brazil | Cross-sectional | Population: primary care; Age: NR; Male n(%): 225 (44.8); Diabetes time: NR; A1c%: | T2DM | Peripheral neuropathy | NSS and NDS: moderate symptoms with/without signs or mild symptoms + moderate signs | Physical exam ≥ two signs + symptoms | Probable | 551 | 35 | 6.3% | 5 |
| 11 | de Souza Lira (2005) [61] | Brazil | Cross-sectional | Population: reference center; Age: 54.2; Male n(%): 43 (38.1); Diabetes time: Debut; A1c%: NR | T2DM | Peripheral neuropathy | ≥1/3: Achilles reflex, vibration 128 Hz tuning-fork test, monofilament. | Physical exam ≥ two signs | Possible | 113 | 29 | 25.7% | 7 |
| 12 | Del Brutto (2016) [62] | Ecuador | Cross-sectional | Population: general population; Age: 64; Male n(%): 51 (46.4); Diabetes time: NR; A1c%: NR | T2DM | Diabetes complications | MNSI symptom >7, MNSI physic exam ≥ 2.5/10 | Physical exam ≥ two signs + symptoms | Probable | 110 | 65 | 59.0% | 8 |

(Continued)

**Table 1.** (Continued)

| | Author (year) | Country | Design | Population, age (mean-years), male (%), diabetes time (median-years), A1c % (mean) | DM type | Primary outcome | Diagnostic criteria according to the study | Grouping criteria | Toronto Diabetic Neuropathy Expert Group Criteria | Sample size (N) | Diabetic neuropathy cases (n) | Prevalence (%) | Quality assessment (total score) |
|---|---|---|---|---|---|---|---|---|---|---|---|---|---|
| 13 | Di Lorenzo (2020) [63] | Uruguay | Cross-sectional | Population: Reference center. Age: NR. Male n(%): 36 (44.4). Daibetes time: NR. A1c%: | T1DM y T2DM | Peripheral neuropathy | TSS and NDS: moderate/severe signs with/without symptoms or mild signs with symptoms | Physical exam ≥ two signs + symptoms | Probable | 81 | 28 | 34.6% | 4 |
| 14 | Dutra (2018) [64] | Brazil | Cross-sectional | Population: reference center. Age:50.8. Male n(%): NR. Diabetes time: 12.5. A1c%: 8.25 | T1DM/T2DM | Diabetes complications | Symptom Achilles reflex, vibration, temperature, pain perception | Physical exam ≥ two signs + symptoms | Probable | 117 | 68 | 58.1% | 4 |
| 15 | Ferreira (2005) [65] | Brazil | Cross-sectional | Population: reference center. Age: 12.9. Male n(%): 28 (58.3). Diabetes time: 6. A1c%: NR | T1DM | Peripheral neuropathy | Nerve conduction test | Nerve conduction test | Confirmed/Sub clinic | 48 | 29 | 60.4% | 4 |
| 16 | Gerchman (2008) [66] | Brazil | Cross-sectional | Population: reference center. Age: 58.5. Male n(%): 848 (46.9). Diabetes time: 11.9. A1c%: 7.2 | T2DM | Diabetes complications | 2/4: symptoms, Achilles reflex, vibration 128 Hz tuning-fork test, monofilament. | Physical exam ≥ two signs + symptoms | Probable | 1810 | 583 | 32.2% | 7 |
| 17 | Gonzales Milan (2017) [67] | Mexico | Cross-sectional | Population: reference center. Age: 31.4. Male n(%): 13 (27.1). Diabetes time: 12.5. A1c%: NR | T1DM | Peripheral neuropathy | Score ≥1/40: Achilles reflex, ankle strength, vibration 128 Hz tuning-fork test, monofilament, in arms and legs | Physical exam ≥ two signs | Probable | 48 | 35 | 73.0% | 4 |
| 18 | Ibarra (2012) [68] | Mexico | Cross-sectional | Population: primary care. Age: 58. Male n(%): 138 (39.7). Diabetes time: 9. A1c%: NR | T2DM | Peripheral neuropathy | MNSI physic exam ≥ 2/10 | Physical exam ≥ two signs + symptoms | Probable | 348 | 240 | 69.0% | 8 |

(Continued)

Table 1. (Continued)

| | Author (year) | Country | Design | Population, age (mean-years), male (%), diabetes time (median-years), A1c % (mean) | DM type | Primary outcome | Diagnostic criteria according to the study | Grouping criteria | Toronto Diabetic Neuropathy Expert Group Criteria | Sample size (N) | Diabetic neuropathy cases (n) | Prevalence (%) | Quality assessment (total score) |
|---|---|---|---|---|---|---|---|---|---|---|---|---|---|
| 19 | Lazo (2014) [69] | Peru | Cross-sectional | Population: reference center; Age: 59.2; Male n(%): 56 (43.4); Diabetes time: 8.6; A1c%: 8.7 | T2DM | Peripheral neuropathy | DNS + monofilament | Physical exam ≥ two signs + symptoms | Probable | 129 | 73 | 56.6% | 4 |
| 20 | Milan Guerrero (2012) [70] | Mexico | Cross-sectional | Population: reference center; Age: 56.9; Male n(%): 45 (30.0); Diabetes time: 8; A1c%: NR | T2DM | Peripheral neuropathy | 2 criteria: MNSI ≥2/10 y Nerve conduction test | Physical exam ≥ two signs + nerve conduction test | Confirmed/Sub clinic | 150 | 131 | 87.3% | 4 |
| 21 | Moreira (2009) [71] | Brazil | Cross-sectional | Population: general population; Age: 56.2; Male n(%): 68 (31.8); Diabetes time: NR; A1c%: NR | T2DM | Peripheral neuropathy | NSS and NDS | Physical exam ≥ two signs + symptoms | Probable | 214 | 39 | 19.1% | 4 |
| 22 | Moreira (2007) [72] | Brazil | Cross-sectional | Population: reference center; Age: NR; Male n(%): 12 (18.5); Diabetes time: 9.88; A1c%: NR | T2DM | Peripheral neuropathy | NSS and NDS | Physical exam ≥ two signs + symptoms | Probable | 65 | 22 | 33.8% | 4 |
| 23 | Paisey (1984) [73] | Mexico | Cross-sectional | Population: reference center; Age: 52.2; Male n(%): 199 (39.6); Diabetes time: 10.7; A1c%: NR | T2DM | Diabetes complications | Signs with or without symptoms: Achilles reflex, vibration in ankle | Physical exam ≥ two signs + symptoms | Probable | 503 | 205 | 40.8% | 8 |
| 24 | Rivas (2016) [74] | Mexico | Cross-sectional | Population: reference center; Age: 56.4; Male n(%): 59 (29.8); Diabetes time: 12.3; A1c%: NR | T2DM | Peripheral neuropathy | MNSI physic exam ≥ 2/10 | Physical exam ≥ two signs | Probable | 198 | 130 | 65.70% | 3 |

(Continued)

**Table 1.** (Continued)

| | Author (year) | Country | Design | Population, age (mean-years), male (%), diabetes time (median-years), A1c % (mean) | DM type | Primary outcome | Diagnostic criteria according to the study | Grouping criteria | Toronto Diabetic Neuropathy Expert Group Criteria | Sample size (N) | Diabetic neuropathy cases (n) | Prevalence (%) | Quality assessment (total score) |
|---|---|---|---|---|---|---|---|---|---|---|---|---|---|
| 25 | Rodriguez (2018) [25] | Peru | Cross-sectional | Population: general population; Age: NR; Male n(%): 122 (40.5); Diabetes time: NR; A1c%: NR | T2DM | Diabetes complications | Monofilament with or without tuning-fork test | Physical exam ≥ two signs | Possible | 301 | 40 | 13.30% | 3 |
| 26 | Scheffel (2004) [76] | Brazil | Cross-sectional | Population: reference center; Age: 59; Male n(%): 390 (55.9); Diabetes time: NR; A1c%: 6.8 | T2DM | Diabetes complications | Symptoms + 1/3 physical exam: Achilles reflex, vibration 128 Hz tuning-fork test, monofilament | Physical exam ≥ two signs + symptoms | Probable | 698 | 251 | 36.0% | 7 |
| 27 | Ticse (2013) [77] | Peru | Cross-sectional | Population: reference center; Age: 57.7; Male n(%): 17 (27.4); Diabetes time: 7.8; A1c%: 9.6 | T2DM | Peripheral neuropathy | Nerve conduction test | Nerve conduction test | Confirmed/Sub clinic | 62 | 60 | 96.7% | 6 |
| 28 | Tres (2007) [78] | Brazil | Cross-sectional | Population: reference center; Age: 57.8; Male n(%): 137 (40.3); Diabetes time: 8; A1c%: 8.1 | T2DM | Peripheral neuropathy | ≥ 3/6 physical exam: monofilament, tuning-fork test, temperature, Achilles reflex, muscular strength, pinprick test. | Physical exam ≥ two signs | Probable | 340 | 75 | 22.0% | 5 |

NR, Not reported; DM, Diabetes Mellitus; T1DM, Type 1 Diabetes Mellitus; T2DM, Type 2 Diabetes Mellitus; TCNS, Toronto Clinical Neuropathy Score; MNSI, Michigan Neuropathy Screening Instrument; NSS, Neuropathy Symptoms Score; NDS, Neuropathy Disability Score; PCN, Partial Constriction Neuropathy; MDNS, Michigan Diabetic Neuropathy Score. NCT: Nerve Conduction Test.

**Table 2. Characteristics of two included studies of diabetic peripheral neuropathy incidence in Latin-American and the Caribbean countries.**

| Author year | Country | Design | Type population, age (mean-years), male (%), diabetes time (media-years), A1c % (mean), | DM type | Main outcome | Basal diagnostic criteria | Follow-up diagnostic criteria. | Grouping criteria | Toronto Diabetic Neuropathy Expert Group Criteria | Basal sample size (N) | Diabetic neuropathy cases (n) | Follow time months (median) | Incidence (%) | Quality assessment NCO (total score) |
|---|---|---|---|---|---|---|---|---|---|---|---|---|---|---|
| Cardoso (2008) [56] | Brazil | Cohort | Population: reference center / Age: 60.5 / Male n(%): 250 (53.1) / Diabetes time: 9.3 / A1c%: NR | T2DM | Diabetes complications | $\geq$ 2/4: symptoms, monofilament, tuning-fork test, altered reflexes | Development | Physical exam $\geq$ 2 signs + symptoms | Probable | 403 | 48 | 57 | 11.9% | 6 |
| Massardo (2019) [79] | Chile | Cohort | Population: reference center / Age: NR / Male n(%): NR / Diabetes time: NR / A1c%: 8.7 | T2DM | Diabetes complications | MNSI symptom >4/ 10, MNSI physic exam $\geq$ 2/10 | Development | Physical exam $\geq$ two signs | Probable | 32 | 17 | 119.3 | 54.8% | 6 |

T1DM, Type 1 Diabetes Mellitus; T2DM, Type 2 Diabetes Mellitus; MNSI, Michigan Neuropathy Screening Instrument

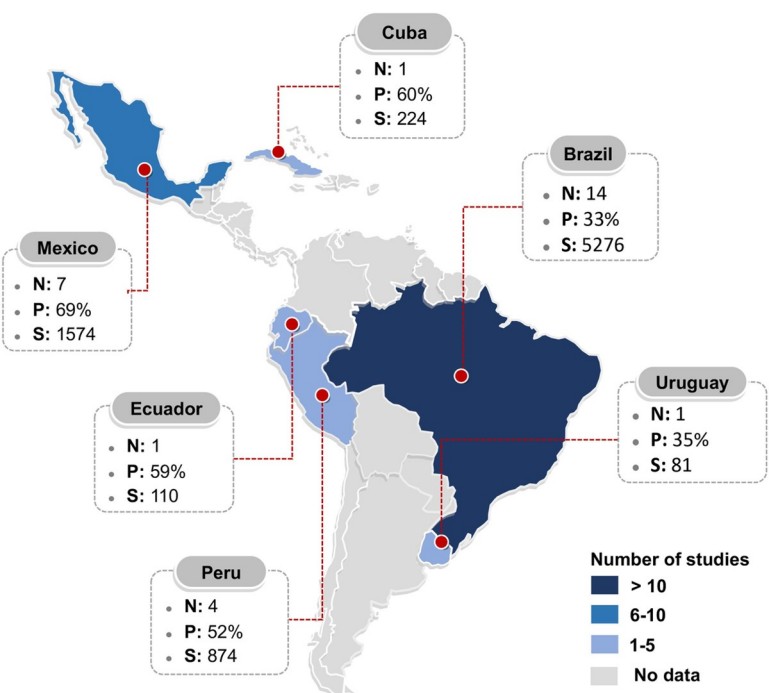

**Fig 2. Prevalence of DPN in LAC: Characteristics and geographic location of included studies.** N, number of studies; P, pooled prevalence (percentage); S, cumulative sample size. DPN: Diabetic peripheral neuropathy. LAC: Latin America and the Caribbean.

studies reported point prevalence data. Only two studies from Brazil and Chile reported incidence estimates, and we found DPN incidence of 13.7 (95% CI: 10.6–17.2) $I^2$: Not calculated [56, 79] (**S1 Fig**).

The cumulative meta-analysis revealed that the pooled estimate changed over time as each study is added to the pool. We found an overall positive trend of the DPN prevalence and reduction of the estimation precision, from 40.7% (95% CI: 32.0 to 49.5) in 1984 to 45.8 (95% CI: 38 to 54) in 2020. The smallest pooled estimate was 29.6% (95% CI: 23 to 36) in 2009 and the highest was 49.5% (95% CI: 40% to 58%) in 2018 (**Fig 4**).

## Subgroup analysis

We stratified the studies according to country, age group, population, number of health centers, diabetes type, diabetes time, primary outcome, and DPN Toronto criteria. Mexico and Cuba had the highest estimated prevalence of 68,7% (95% CI: 55.1 to 80.8) and 60,2% (95% CI: 53.5 to 66.7), respectively; while Brazil had the lowest estimated prevalence (33.1%, 95% CI: 24.8to 40.8) (**S2 Fig**). The subgroup analysis for the type of DM showed no differences (heterogeneity test between-groups, p = 0.53) among patients with type 1 DM (54.8%, 95% CI: 30.8 to 77.7) compared to type 2 DM (44.8%, 95% CI: 35.4 to 54.4). Also, the DPN prevalence was higher (heterogeneity test between-groups, p<0.001) when the duration of DM was greater than ten years (72,9%; 95% CI: 58.1 to 84.7) compared to debut DM 25.7% (95% CI: 17.9 to 34.7). According to DPN Toronto criteria subgroups, the subclinical DPN had higher prevalence estimates (78,8%; 95% CI: 57.8 to 94.0, heterogeneity test between-groups, p<0.001) compared to possible DPN (23.8%; 95% CI: 11.1 to 39.5) (**S3 Fig**). There were no significant differences in the estimates of DPN prevalence according to age group and number of health centers included in **Table 3**.

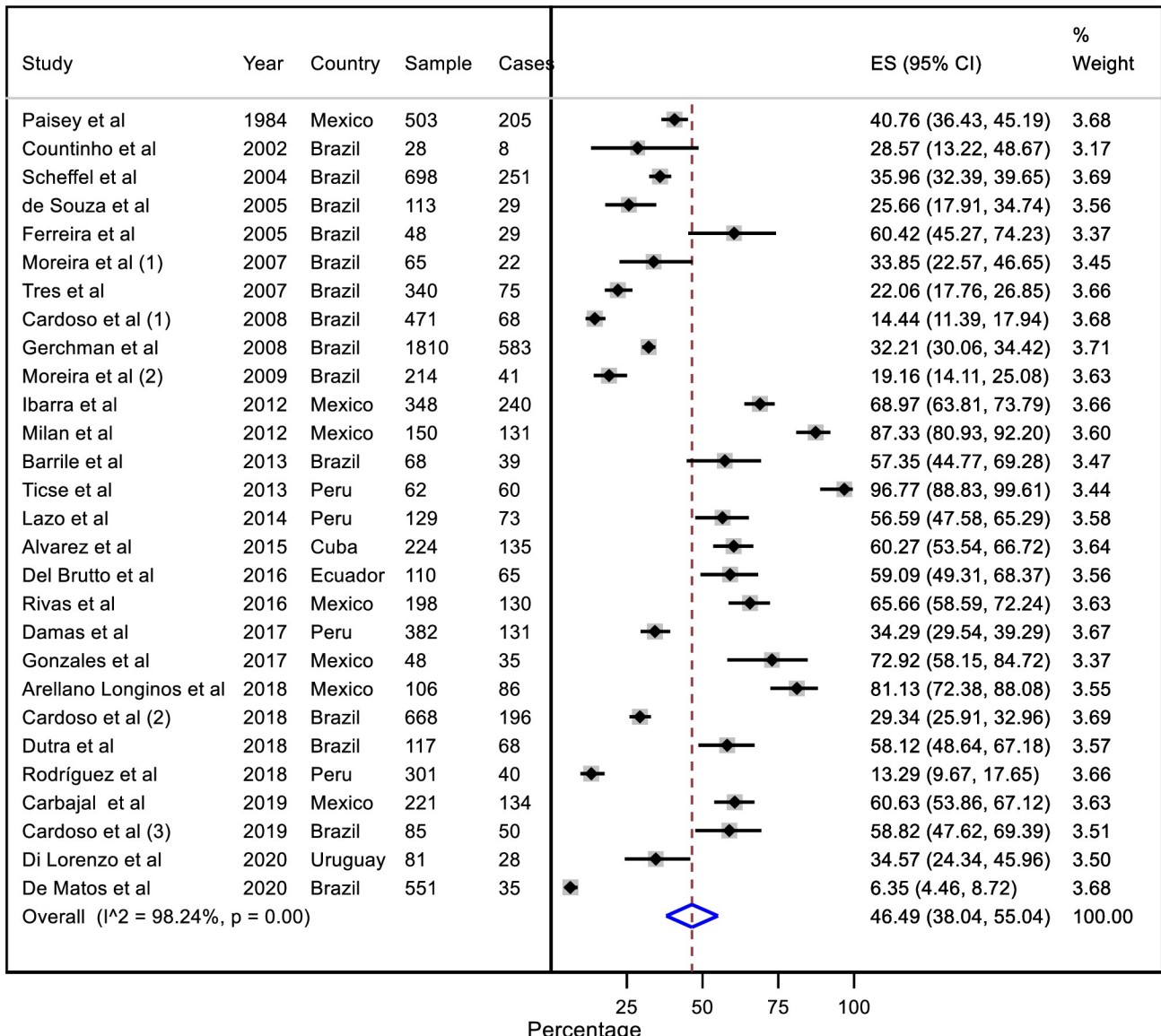

**Fig 3. Forest plot (random-effects model) of a meta-analysis of diabetic peripheral neuropathy prevalence in Latin America and the Caribbean countries.**

## Sensitivity analysis

By removing individual studies excluding each study, the pooled prevalence of DPN varied from 46.4% (95% CI: 38.3 to 54.5) to 50.0% (95% CI: 40.9 to 59.1). Our analysis shows no influence of a single study on the pooled estimate's direction or magnitude (**Fig 5**). The study's quality revealed that the DPN prevalence was higher in moderate-quality studies 49.6% (95% CI: 26.1 to 73.28), while for high-quality studies was lower 42,1% (95% CI: 32.6 to 52.0). The results were similar for studies with a sample size of less than 323; higher prevalence was found in studies with a small sample size (54.8%; 95% CI: 42.5 to 66.9) compared to large sample size (30.3%; 95% CI: 20.5 to 41.1) (**Table 4**).

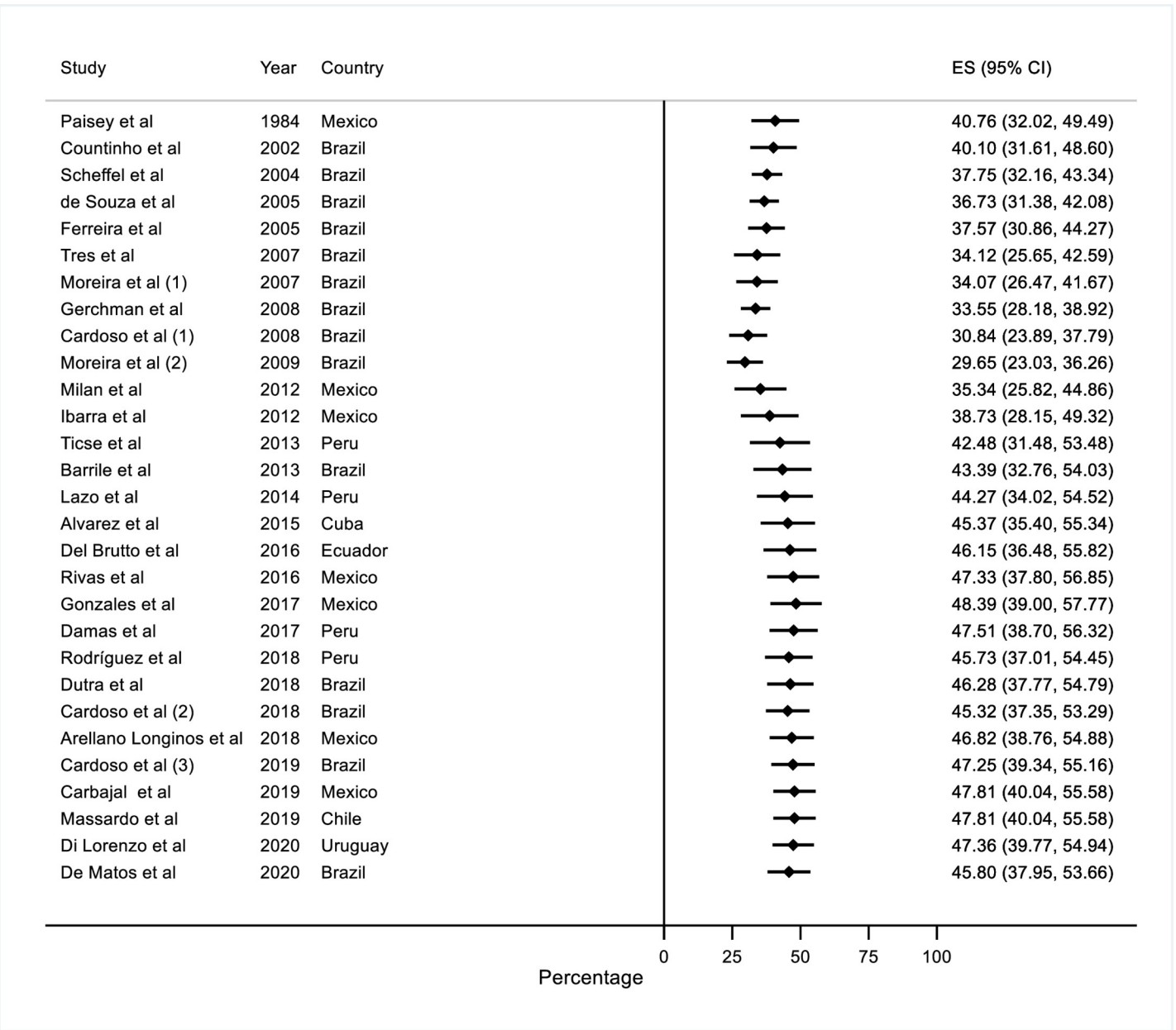

**Fig 4. Cumulative meta-analysis of diabetic peripheral neuropathy prevalence in Latin America and the Caribbean countries.**

## Meta-regression analysis

The univariate meta-regression models showed significant association of A1c hemoglobin (%) with DPN prevalence (b = 0.2; 95% CI: 0.01 to 0.37; p = 0.04) and explained 49.69% (by adjusted $R^2$) of the variance. This value represents a prevalence increase of 2% by one unit increase of A1c hemoglobin in the included studies. Besides, the studies with DPN prevalence as primary outcome (no focusing on broad range of DM complications) reported lower estimates (b = -0.12; 95% CI: -0.30 to 0.06; p = 0.18; $R^2$ = 3.85%), and the studies with subclinical DPN patients (based on Toronto criteria) reported higher prevalence (b = 0.51; 95% CI: 0.19 to 0.84; p = 0.003; $R^2$ = 32.8%). There was no association with age, time of diabetes, year of publication, sample size, and quality score (**Table 5**). The multivariate model with Monte-

**Table 3. Subgroup analysis of meta-analysis of diabetic peripheral neuropathy prevalence in Latin America and the Caribbean countries.**

| | N | Prevalence | 95% CI | % weight | I2 |
|---|---|---|---|---|---|
| **Country** | | | | | |
| Cuba | 1 | 60.26 | 53.53–66.72 | 3.64 | . |
| Mexico | 7 | 68.70 | 55.12–80.83 | 25.12 | 96.58 |
| Brazil | 14 | 33.13 | 24.87–40.82 | 49.83 | 96.97 |
| Peru | 4 | 51.60 | 21.52–81.06 | 14.35 | 98.76 |
| Ecuador | 1 | 59.09 | 49.31–68.37 | 3.56 | . |
| Uruguay | 1 | 34.57 | 24.34–45.96 | 3.50 | 98.24 |
| **Type of population** | | | | | |
| General population | 3 | 28.64 | 8.75–54.25 | 10.84 | . |
| Primary care | 4 | 51.67 | 10.75–91.26 | 14.36 | 99.49 |
| Reference center | 21 | 48.15 | 40.02–56.32 | 74.79 | 97.45 |
| **Type of DM** | | | | | |
| Type 1 Diabetes Mellitus | 3 | 54.85 | 30.86–77.75 | 9.91 | . |
| Type 2 Diabetes Mellitus | 23 | 44.87 | 35.48–54.44 | 79.39 | 98.51 |
| Both | 3 | 51.48 | 37.03–65.80 | 10.70 | . |
| **Age group** | | | | | |
| < 18 years old | 2 | 48.53 | 37.19–59.94 | 6.54 | . |
| ≥18 years old | 26 | 46.61 | 37.87–55.46 | 93.46 | 98.35 |
| **Time of DM** | | | | | |
| Debut of DM | 1 | 25.66 | 17.91–34.73 | 3.56 | . |
| >5 years of DM | 6 | 44.99 | 21.87–69.287 | 21.14 | 99.15 |
| > 10 years of DM | 1 | 72.91 | 58.15–84.72 | 3.37 | . |
| Any time of DM | 20 | 46.74 | 37.24–56.36 | 71.93 | 97.85 |
| **Primary outcome** | | | | | |
| Diabetes neuropathy | 17 | 51.91 | 35.83–67.80 | 67.80 | 97.00 |
| DM complication | 11 | 38.31 | 30.49–46.44 | 40.04 | 98.59 |
| **Type of diagnostic** | | | | | |
| Clinical signs | 9 | 47.43 | 31.32–63.82 | 32.24 | 97.93 |
| Clinical signs + symptoms | 14 | 37.67 | 28.23–47.60 | 50.54 | 98.15 |
| Clinical signs and NCT | 1 | 28.57 | 13.22–48.66 | 3.17 | . |
| Only NCT or SDT | 4 | 78.80 | 57.83–94.03 | 14.05 | 95.53 |
| **Type of peripheral neuropathy according to Toronto criteria** | | | | | |
| Possible DPN | 3 | 23.82 | 11.10–39.49 | 10.88 | . |
| Probable DPN | 20 | 44.25 | 35.20–53.49 | 71.90 | 98.18 |
| Confirmed DPN | 1 | 28.57 | 13.22–48.66 | 3.17 | . |
| Subclinical +Confirmed DPN | 4 | 78.80 | 57.83–94.03 | 14.05 | 95.53 |
| **Number of health centers** | | | | | |
| Single center | 25 | 47.16 | 36.61–57.85 | 89.04 | 98.39 |
| Multicenter | 3 | 40.58 | 31.47–50.03 | 10.96 | . |

DM, Diabetes Mellitus; DPN, Diabetic Peripheric Neuropathy; NCT, Nerve Conduction Tests; SDT, Sudomotor Dysfunction Tests.

Carlo permutations showed a significant association of the subclinical DPN category, namely, the studies with patients in this category reported higher estimates (b = 0.46; 95% CI: 0.13 to 0.79; p = 0.008) compared to the others definitions, adjusted by sample size, and studies with DPN as the primary outcome. The multivariate model explained 38.1% of the prevalence

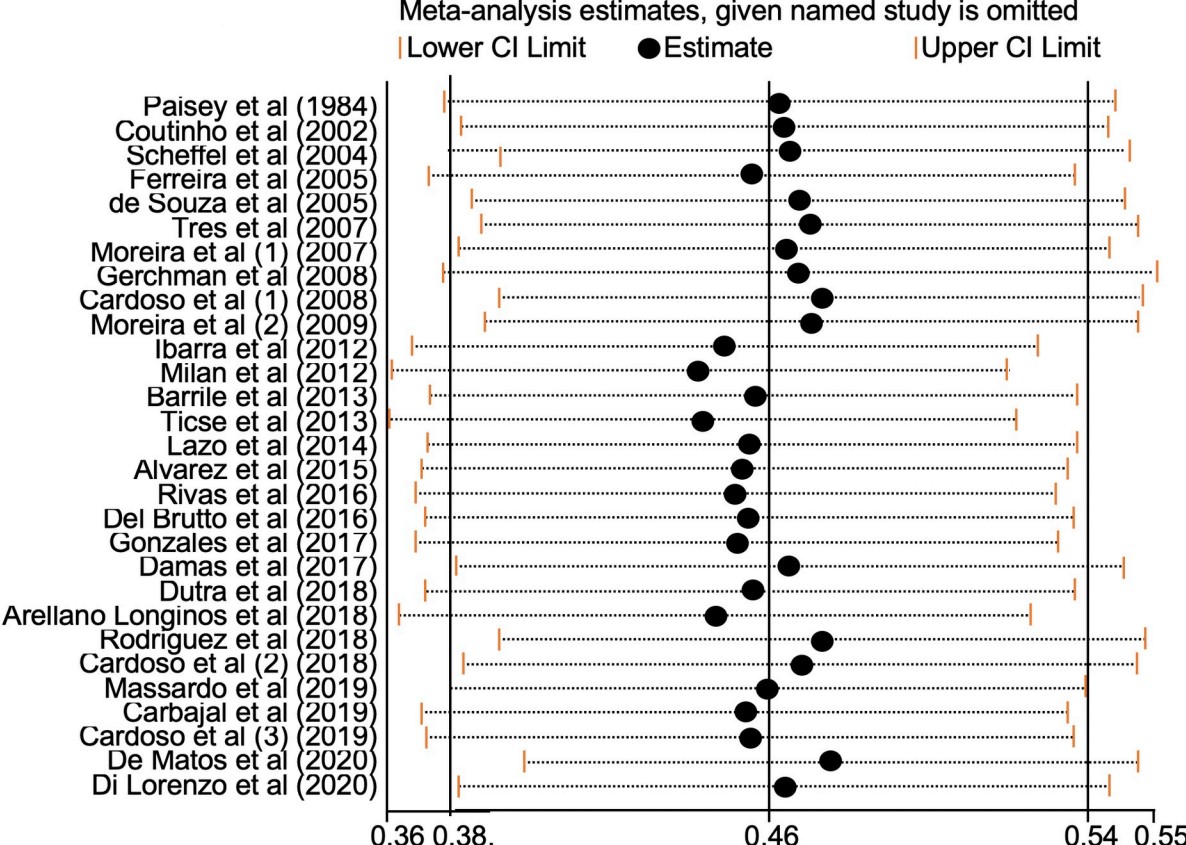

**Fig 5. Sensitivity analyses through consecutively excluding the 28 included studies.**

variance. Unfortunately, we could not include A1c hemoglobin in this model due to the small number of studies reporting this value.

## Publication bias

The visual inspection of the funnel plot showed asymmetrical distribution, which indicated the presence of publication bias. (**Fig 6A**) This finding was corroborated by Egger's test (p = 0.010). From the trim and fill model, nine studies were imputed based on a linear trimming estimator (**Fig 6B**), the filled random-effects meta-analysis calculated a pooled prevalence of 31.1% (95% CI: 22.6 to 39.5).

## Evidence certainty

We judged the certainty of the available evidence as very low. We started the evaluation with low certainty because only three population-based studies were included. We downgraded, according to the high risk of bias of the included studies (46% of the studies had a low and very-low quality by Loney's scale). Besides, we downgraded the evidence body due to publication bias and high inconsistency ($I^2$ was >60%) in the meta-analysis (**Table 6**).

## Discussion

In this systematic review and meta-analysis, the overall prevalence of DPN in LAC was 46.5% (95%CI: 38.0 to 55.0; $I^2$ = 98.2%, p < 0.01), and the incidence (from two studies) was 13.7%

**Table 4. Sensitivity analysis of meta-analysis of diabetic peripheral neuropathy prevalence in Latin American and the Caribbean countries.**

| | N | Prevalence | 95% CI | % weight | I2 |
|---|---|---|---|---|---|
| **Type of sampling** | | | | | |
| Randomized | 10 | 54.18 | 42.83–65.32 | 36.15 | 97.95 |
| No randomized | 18 | 42.10 | 30.03–54.67 | 63.85 | 98.27 |
| **Type of recollection** | | | | | |
| Retrospective | 7 | 36.03 | 30.52–41.74 | 25.47 | 90.85 |
| Prospective | 21 | 50.09 | 36.92–63.24 | 74.53 | 98.62 |
| **Blind evaluation** | | | | | |
| No blind evaluation | 11 | 42.18 | 29.82–55.05 | 39.70 | 98.47 |
| Blind evaluation | 17 | 49.37 | 37.37–61.40 | 60.30 | 98.05 |
| **Sample ≥ 323 subjects** | | | | | |
| No | 19 | 54.84 | 42.49–66.90 | 66.87 | 97.22 |
| Yes | 9 | 30.33 | 20.52–41.13 | 33.13 | 98.61 |
| **Size sample according to precision** | | | | | |
| <165 (precision >7%) | 14 | 59.44 | 46.78–71.50 | 48.70 | 94.87 |
| 165–322 (precision 5–7%) | 5 | 42.50 | 19.97–66.81 | 18.18 | 98.62 |
| 323–800 (precision 3–5%) | 8 | 30.10 | 18.07–43.70 | 29.41 | 98.77 |
| >896 (precision <3%) | 1 | 32.21 | 30.06–34.41 | 3.71 | . |
| **Quality clinic study** | | | | | |
| Very low (0–2) | 2 | 44.84 | 36.88–52.94 | 6.97 | . |
| Low (3–4) | 14 | 47.86 | 30.80–65.17 | 49.48 | 98.59 |
| Moderate (5–6) | 5 | 49.64 | 26.09–73.28 | 18.02 | 98.88 |
| High (7–8) | 7 | 42.16 | 32.63–51.98 | 25.52 | 96.96 |

[a] Adequate sample size > 323 subjects, considering the prevalence of DPN of 30% according to Sun et al. [27], 5% alpha, and 80% of power.

(95% CI: 10.6 to 17.2; $I^2$ = Not calculated). We found an increasing trend of cumulative DPN prevalence over time. The main sources of heterogeneity associated with higher prevalence were: i) factors related to the included population (the diagnosis criteria such as subclinical DPN, the higher duration of diabetes mellitus, and the higher percentage of glycosylated hemoglobin); and factors related to the methodology of included studies (high risk of bias and sample size). Based on the high heterogeneity, high risk of bias, and publication bias, we judge the included evidence as very low certainty.

Our results showed that the prevalence of DPN in LAC is similar to the estimates from Iran and Africa, with 53% (95% CI: 41 to 65) and 46% (95% CI: 36.21 to 55.78), respectively [25, 26]. Nonetheless, our finding is higher than the global prevalence, which was estimated as 35.78% (95% CI: 27.86 to 44.55; $I^2$ = 99%, p <0.001) and 30% (95% CI: 25 to 34; $I^2$ = 99.5%, p <0.001) [80, 27]. Additionally, we found a positive trend of cumulative DPN prevalence over time from 1984 to 2019. This difference and increasing pattern could be explained by better prevention strategies implemented in developed countries [26]. It is worth mentioning that Sun *et al.* meta-analysis did not include studies from LAC, while Souza *et al.* only included two studies from this region [27, 80]. Therefore, to our knowledge, this is the first study synthesizing the DPN epidemiological estimates from LAC.

Published data of well-designed studies about DPN incidence are limited in the world, especially in LAC [81]. We included two studies that reported DPN incidence in T2DM, from Brazil and Chile (11.9% and 54.8%, respectively) [53, 79]. Compared to a prospective study

**Table 5. Meta-regression models of diabetic peripheral neuropathy prevalence in Latin-American and the Caribbean countries.**

| | | Crude | | | | | Adjusted Model [a] | | |
|---|---|---|---|---|---|---|---|---|---|
| | n | B | 95% CI | P value | I$^2$ | Adjusted R$^2$ | β | 95% CI | P value |
| **Age (years)** | 23 | -0.003 | -0.01 to 0.0061 | 0.43 | 90.3 | -0.88 | | | |
| **Male (%)** | 24 | -0.003 | -0.01 to 0.05 | 0.46 | 90.4 | -2.0 | | | |
| **Diabetes time (years)** | 15 | -0.007 | -0.06 to 0.05 | 0.79 | 92.5 | -7.43 | | | |
| **A1c hemoglobin (%)** | 8 | 0.19 | 0.01 to 0.37 | 0.04 | 84.1 | 49.6 | | | |
| **Year of publication** | 28 | 0.005 | -0.006 to 0.17 | 0.36 | 91.53 | -1.5 | | | |
| **Quality score** | 28 | 0.007 | -0.065 to 0.049 | 0.77 | 91.5 | -4.5 | | | |
| **Sample size (≥ 323) [b]** | 28 | -0.22 | -0.39 to -0.052 | 0.01 | 89.18 | 21.7 | -0.19 | -0.36 to -0.01 | 0.03 |
| **DPN as primary outcome** | 28 | -0.12 | -0.30 to 0.06 | 0.18 | 91.02 | 3.85 | 0.02 | -0.15 to 0.19 | 0.78 |
| **DPN Toronto criteria** | 28 | | | | 87.9 | 32.8 | | | |
| **Possible** | | 1 | | | | | 1 | | |
| **Probable** | | 0.19 | -0.05 to 0.45 | 0.13 | | | 0.21 | -0.02 to 0.46 | 0.26 |
| **Confirmed** | | 0.04 | -0.55 to 0.63 | 0.88 | | | -0.01 | -0.59 to 0.58 | 0.97 |
| **Confirmed-subclinical** | | 0.51 | 0.19 to 0.84 | 0.003 | | | 0.46 | 0.13 to 0.79 | 0.008 |

[a]Adjusted by sample size, Toronto Criteria and DPN as primary outcome. Adjusted R$^2$ = 38.1; I$^2$ = 88.4; p = 0.012.
[b] Adequate sample size > 323 subjects, considering the prevalence of DPN of 30% according to Sun et al. [27], 5% alpha, and 80% of power

conducted on the American population with T1DM, our results are significantly lower (13.7% versus 30%) [82]. However, these results are not comparable due to the different types of diabetes mellitus included in both studies. Therefore, more population-based cohorts are needed to determine the incidence of DPN in LAC.

## Sources of heterogeneity

The prevalence of DPN varied from 13% to 97% in the included studies. This values aligned with Sobhani *et al.* and Shiferaw *et al.* meta-analyses, who attributed this variation to the different diagnostic criteria of DPN used in the studies [25, 26]. In this review, we used the Toronto Diabetic Neuropathy Expert group's definitions to evaluate DPN diagnosis, which classified DPN as confirmed (abnormal nerve conduction and a symptom or sign of neuropathy); probable (a combination of symptoms and signs of neuropathy); possible DPN (any symptoms or signs); and subclinical (no signs or symptoms of neuropathy with abnormal nerve conduction test) [35]. Interestingly, we found that the highest pooled DPN prevalence was observed from studies that used subclinical DPN criteria (78.8%; 95% CI: 57.83 to 94.03), and the lowest was observed with the possible DPN criteria (23.8%; 95%CI: 11.1 to 39.5). This disparity could result from a more precise diagnostic accuracy of the nerve conduction test, which is a requirement for subclinical criteria and could lead to a decrease in the likelihood of false-positive test results.

Besides, our multivariate meta-regression model results confirm that the main source of DPN prevalence heterogeneity is the used diagnostic criteria. This finding is consistent with Sun *et al.* [27], who found that unclear diagnosis was the only criteria associated with heterogeneity for DPN, obtaining an odds ratio of 2.92 (95% CI 1.08 to 1.25; p <0.05). Taking this into account, we excluded 11 studies that used an inadequate diagnostic method, such as only

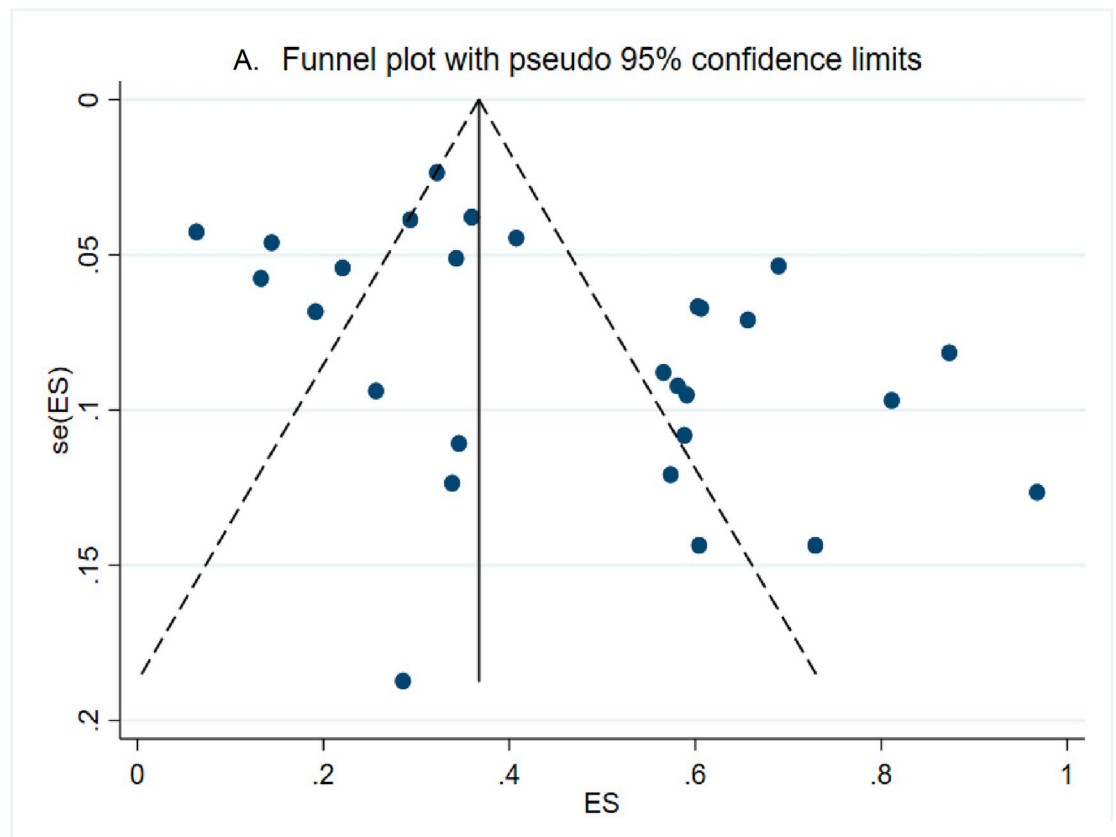

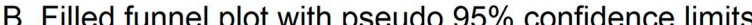

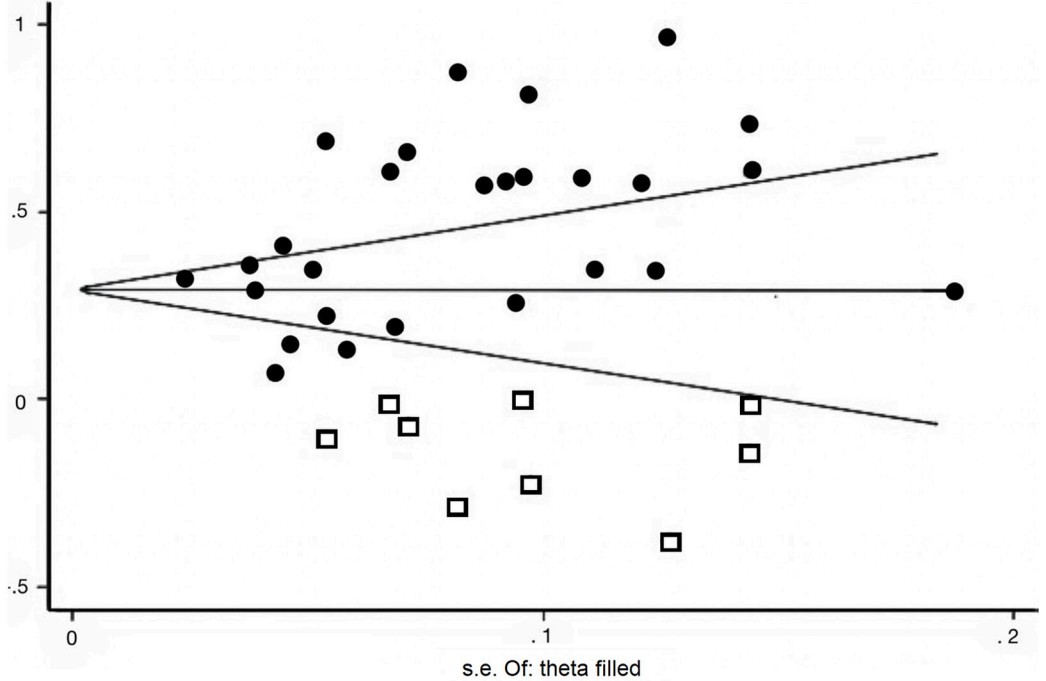

**Fig 6. Funnel plot of the overall prevalence of DPN in the 25 included studies.** A. Classic Funnel Plot. B. Funnel plot with trimmed studies.

extracting the diagnosis from medical records, self-report diagnosis, and clinical examination without specifying which method was used. Moreover, we excluded nine studies with insufficient data on the diagnosis criteria. For instance, we excluded studies that relied on one sign in the physical examination and reported pain. Despite our efforts to clarify the diagnostic criteria and case definition, we also obtained a high heterogeneity associated with diagnostic criteria, although our pooled estimates could be considered more accurate than previous meta-analyses.

Additionally, the estimated DPN prevalence was higher in Mexico and Cuba, while Bra+zil had the lowest estimated prevalence. Poor glycemic control is shared among the LAC population and is illustrated in a study conducted in nine Latin American Countries, in which Mexico had a worse glycemic control than Brazil, measured by HbA1c $> 7$%, and this marker increased significantly with a longer duration of T2DM [83]; therefore, it is a significant risk factor for DPN [84]. These findings are also aligned with our subgroup and univariate meta-regression results showing that the duration of DM and percentage of glycosylated hemoglobin were associated with higher DPN prevalence. It is well-know that uncontrolled hyperglycemia (due to a poor glycemic control) leads to activation of different mechanisms, such as the polyol pathway, generation of AGEs (advanced glycation end-products) and ROS (reactive oxygen species), and activation of the protein kinase C (PKC) pathway [85]. These mechanisms play a significant role in the pathogenesis of DPN and exacerbate with worse glycemic control and longer duration of disease [86].

Regarding the DM type, due to the small number of studies on T1DM and its wide confidence interval, it is not possible to establish differences in prevalence between both types. Nonetheless, in a recent meta-analysis by Sun *et al.* [27], they found that patients with type

**Table 6. Quality of the body of evidence according to GRADE: Summary of findings.**

| Outcomes | Anticipated absolute effects (95% CI) | | № of participants | The certainty of the evidence |
|---|---|---|---|---|
| | **Frequency pooled (%)** | **CI 95%** | **(Studies)** | **(GRADE)** |
| Prevalence of diabetic peripheral neuropathy in LAC | 46.5 | 38.0 to 55.0 | 8139 | ⊕◯◯◯ |
| | | | (28 studies) | VERY LOW [a, b, c, d, e] |
| Incidence of diabetic peripheral neuropathy in LAC | 13.7 | 10.6 to 17.2 | 503 | ⊕◯◯◯ |
| | | | (2 studies) | VERY LOW [f, g,h] |

CI, Confidence interval; LAC, Latin America and the Caribbean.

For prevalence

[a] The certainty rating started from low certainty since only three population-based studies were included

[b] High risk of bias (low and very low quality by Loney's scale) was detected in most of the included studies (52%), due to the inadequate sample size, sampling, and evaluation.

[c] High inconsistency was detected in meta-analyses. The calculated $I^2$ was $>60$%.

[d] Publication bias was detected in the meta-analysis by the funnel plot and Egger's test.

[e] Not imprecision by adequate sample size and narrow confidence interval.

For incidence

[f] The certainty rating started from low certainty since only two population-based study was included.

[g] Risk of bias (moderate-quality by New Castle-Ottawa scale) was detected in both included study.

[h] We do not evaluate inconsistency, publication bias, or imprecision for this outcome.

T2DM presented a higher DPN prevalence than those with type T1DM. This disparity may be explained by the differences in the pathophysiology of DPN between both types. In T2DM, dyslipidemia, insulin resistance, and systemic inflammation are significant factors for DPN, which can develop before diabetes onset and diagnosis [16]. In contrast, T1DM onset is mostly correlated with the presentation of symptoms, caused primarily by insulin deficiency and hyperglycemia, which is why a tight glucose control could reduce the risk of DPN in T1DM, but not in T2DM [87].

Since we decided to included studies with mixed populations (T1DM or/and T2DM) in our overall pooled estimates in order to increase generalizability and to reduce the impact of potential diagnosis overlap between T1DM and T2DM, we performed a sensitivity analysis to explore heterogeneity, that showed non-important impact of T1DM population into the regional pooled estimates (likely owing to the small number of included studies and sample size), contrary to previous studies showing less prevalence of DPN in T1DM [81]. However, it is important to considering that current studies postulate that the diabetes mellitus classification is insufficient, and it has been suggested 5 phenotypes that would better explain the long-term results [88]. Moreover, the Latent Autoimmune Diabetes in Adults (LADA) could simulates T2DM onset and turns to a total insulin deficiency in a short-term period [89], thus complicating the DM type differentiation Additionally, the unavailability of laboratory tests for T1DM-specific antibodies in LAC countries could hamper the correct diagnosis [32]. Therefore, it is important to increase the number and quality of the T1DM diabetes registries in LAC, to estimate a precise DPN prevalence in this population.

We found a significant methodological issue in the included papers. Less than half of the studies are considered moderate to high quality. Although, all the studies used a validated criterion for DNP screening and confirmation. Only ten used random sampling, 17 used a blind assessor, and only eight studies included the minimal sample size (323 DM patients) to detect the global prevalence (30%). Finally, only three studies recruited participants from population-based sources; therefore, our estimates are highly influenced by hospital-based participants. We reaffirmed the critical role of study quality in our subgroups and meta-regression analyses, the studies with low quality, small sample size, and those studies with different primary outcomes (different than the DPN estimation) reported higher DPN prevalence. Altogether, this methodological gap in LAC studies leads to the certainty of the evidence as very low; hence, we recommend standardization of these procedures in future studies to enhance the confidence on the DPN estimates from LAC and guide the public health interventions in each country.

Although a high heterogeneity is expected in a meta-analysis of prevalence studies [38], we identified several sources of heterogeneity and high variability. Therefore, within-study factors, such as setting (community versus hospital settings), sample characteristics, comorbidities, presence of cardiovascular risk factors, glycemic control, and used treatments, should be explored in future studies to understand the remaining variability. New longitudinal studies are needed to allow better comparisons between subgroups with specific characteristics over time.

## Limitations and strengths

The present study has some limitations. We only identified studies from five countries, of a total of 33 countries in LAC. Moreover, most of the data was based on a hospital population, with limited general population participation. Therefore, the external validity of our estimates has to be interpreted with caution. Furthermore, although we have identified several heterogeneity sources of the pooled estimates, a large unresolved heterogeneity was found.

Nevertheless, our study has important strengths. It was conducted using a comprehensive search strategy to incorporate all the studies involving LAC patients; therefore, we did not use

any restriction by language or year of publication. Additionally, we used multiple meta-analytic techniques to evaluate the sources of heterogeneity of the DPN estimates. Moreover, we performed an exhaustive quality and certainty of the evidence assessment to identify gaps in the methodology of included studies, further to guide the design of future studies in the region.

## Conclusions

This study revealed that the overall prevalence of DPN was relatively high in LAC countries compared to other regions, as almost half of DM patients presented DPN, although from very low evidence. The significant heterogeneity between and within countries could be explained by population type and methodological aspects. Significant gaps (e.g., under-representation of most countries, lack of incidence studies, and heterogenous case definition) were identified. We suggest DPN should be considered a public health matter in LAC, and health policies should focus on its early detection and prevention to reduce morbidity, impaired quality of life, and the healthcare costs associated with DPN. More population-based studies with better quality are required to evaluate the prevalence of DNP in LAC and its associated factors and the standardization of its evaluation to reduce heterogeneity.

## Supporting information

**S1 Checklist. Prisma checklist of items include reporting a systematic review.**
(DOC)

**S1 Fig. Forest plot of incidence of DPN.**
(TIF)

**S2 Fig. Forest plot according to country of DPN.**
(TIF)

**S3 Fig. Forest plot according to type of DPN according to Toronto criteria.**
(TIF)

**S1 Table. Search strategy.**
(DOCX)

**S2 Table. Studies that were evaluated in full-text, and were excluded.**
(DOCX)

**S3 Table. Quality assessment of prevalence studies.**
(DOCX)

**S4 Table. Quality assessment of incidence studies.**
(DOCX)

## Acknowledgments

We would like to thank to Dr Juan Hiyagon-Kian, Dr César Bonilla-Asalde and Dra Roxana Obando-Zegarra of the Oficina de Apoyo a la Docencia e Investigación (OADI), Hospital Daniel Alcides Carrión for their assistance with the logistic aspects of this study.

## Author Contributions

**Conceptualization:** Marlon Yovera-Aldana, Victor Velásquez-Rimachi, M. D. More-Yupanqui, Fradis Gil-Olivares, César Quispe-Nolazco, Flor Quea-Vélez, Christian Morán-Mariños, Carlos Alva-Diaz.

**Data curation:** Marlon Yovera-Aldana, Mariela Osores-Flores, Fradis Gil-Olivares, César Quispe-Nolazco, Christian Morán-Mariños.

**Formal analysis:** Marlon Yovera-Aldana, Victor Velásquez-Rimachi, Andrely Huerta-Rosario, M. D. More-Yupanqui, Mariela Osores-Flores, Carlos Alva-Diaz, Kevin Pacheco-Barrios.

**Investigation:** M. D. More-Yupanqui, Carlos Alva-Diaz.

**Methodology:** Marlon Yovera-Aldana, Victor Velásquez-Rimachi, Andrely Huerta-Rosario, M. D. More-Yupanqui, Isabel Pinedo-Torres, Carlos Alva-Diaz, Kevin Pacheco-Barrios.

**Project administration:** Marlon Yovera-Aldana.

**Software:** Marlon Yovera-Aldana, Andrely Huerta-Rosario, Ricardo Espinoza, Kevin Pacheco-Barrios.

**Supervision:** Marlon Yovera-Aldana, Victor Velásquez-Rimachi, Isabel Pinedo-Torres, Carlos Alva-Diaz.

**Validation:** Ricardo Espinoza, Christian Morán-Mariños, Isabel Pinedo-Torres, Carlos Alva-Diaz, Kevin Pacheco-Barrios.

**Visualization:** Ricardo Espinoza, Fradis Gil-Olivares, César Quispe-Nolazco, Flor Quea-Vélez, Christian Morán-Mariños, Isabel Pinedo-Torres, Carlos Alva-Diaz, Kevin Pacheco-Barrios.

**Writing – original draft:** Marlon Yovera-Aldana, Victor Velásquez-Rimachi, Andrely Huerta-Rosario, M. D. More-Yupanqui, Mariela Osores-Flores, Ricardo Espinoza, Flor Quea-Vélez, Isabel Pinedo-Torres, Carlos Alva-Diaz, Kevin Pacheco-Barrios.

**Writing – review & editing:** Marlon Yovera-Aldana, Ricardo Espinoza, Carlos Alva-Diaz, Kevin Pacheco-Barrios.

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
