## [Decision Letter · Decision Letter 0]

15 Dec 2020

PONE-D-20-28188

Prevalence and incidence of diabetic peripheral neuropathy in Latin America and the Caribbean: a systematic review and meta-analysis.

PLOS ONE

Dear Dr. Yovera-Aldana,

Thank you for submitting your manuscript to PLOS ONE. After careful consideration, we feel that it has merit but does not fully meet PLOS ONE’s publication criteria as it currently stands. Therefore, we invite you to submit a revised version of the manuscript that addresses the points raised during the review process.

We look forward to receiving your revised manuscript.

Kind regards,

Ahmed Negida, MD

Academic Editor

PLOS ONE

Journal Requirements:

2. Please amend either the title on the online submission form (via Edit Submission) or the title in the manuscript so that they are identical.

Reviewers' comments:

Reviewer's Responses to Questions

**Comments to the Author**

1. Is the manuscript technically sound, and do the data support the conclusions?

Reviewer #1: Yes

Reviewer #2: Yes

Reviewer #3: Yes

2. Has the statistical analysis been performed appropriately and rigorously? 

Reviewer #1: Yes

Reviewer #2: Yes

Reviewer #3: Yes

3. Have the authors made all data underlying the findings in their manuscript fully available?

Reviewer #1: Yes

Reviewer #2: No

Reviewer #3: Yes

4. Is the manuscript presented in an intelligible fashion and written in standard English?

Reviewer #1: No

Reviewer #2: Yes

Reviewer #3: Yes

5. Review Comments to the Author

Reviewer #1: Abstract

1-PubMed is not a database; it is a search engine. You searched MEDLINE via PubMed.

2-Mention the used program for analysis

3-During reporting numbers with fractions or decimals, please use (.) not (,).

4-P-value here 48.5% (95%CI: 40.2-56.8; I2=97.94%; p<0.01) for heterogeneity or effect size?

Introduction

5-Well-written with clear rationale and objectives.

Methods

6- "A systematic review with meta-analysis was performed to assess the prevalence of DPN in patients with DM." Please remove this sentence as it was already mentioned in the objectives.

7- In your protocol, you proposed to identify the prevalence and incidence of "painful" DPN in Latin America, while in the manuscript you aimed to identify the prevalence and incidence of DPN in Latin America and the Caribbean. Please mention this deviation in your manuscript or update the protocol.

8- The inclusion criteria of the population is not clear.

9- It's not clear to me why you excluded studies that reported some comorbidities such as diabetic foot?

10- Databases were searched one year ago, I strongly recommend re-searching the databases for any potential studies.

11- Did you plan to include case reports and case series? If not, please specify.

12- Two independent authors (MMY y MOF). You may mean (MMY and MOF).

13- Line 132: Data >> were not was

14- Please recheck these percentages "quantified using I2 statistical test considering that an I2 < 40% is low, 30-60% is moderate, 50-90% is substantial, and 75-100% is considerable heterogeneity"

15- Authors did not discussed who did they handle the duplicates and missing data.

Results

16- Line 213: There is no supplementary file 8, you may mean S2 Table?!

17- Line 248: It is 48.5% not 48,5%

18- Line 251: 11.9% not 11,9%. This should be addressed in the entire manuscript

19- I could not find S6 Table??!

20- The quality of the figures should be enhanced

21- Is there any explanation for this specific sample size (Sample size (>323) In the meta regression?

Discussion

22- Unresolved heterogeneity should be reported in the limitations.

23- In the conclusion, you have to mention that this is a very low evidence.

General comment

24- Please consider a language editing to eliminate any language mistakes

Reviewer #2: This is an important contribution as it is a systematic review and metanalysis of the incidence and prevalence of DPN in Latin America and the Caribbean, a region with an explosion of diabetes and its complications and as highlighted a real lack of quality data on DPN.

The methods employed for the SR and MA are appropriate.

It highlights the high prevalence of this often neglected complication of diabetes and highlights the lack of rigorous population based studies to define the prevalence of DPN in LAC and the Caribbean.

It also reflects the limited published data from the region and the wide ranging prevalence (13-93%) due to differing populations studied, especially those from secondary care (n=19/25) and the different definitions utilized to diagnose DPN.

Only one small study that assessed the incidence is noted.

It confirms the poor methods used to identify DPN with most studies only being able to identify probable DPN as they rely on symptoms and clinical deficits.

The title should include the word Caribbean.

What about painful DPN?

Reviewer #3: The manuscript is a systematic Review and meta-analysis to estimate the prevalence and incidence of Diabetic Peripheral Neuropathy (PDN) in Latina America and the Caribbean (LAC). The Authors demonstrated that in LAC there is a high (49.5%) prevalence of PDN, a significant health issue. Furthermore, the Authors identify significant heterogeneity between and within country. The results are convincing and indicated several gaps, including under-representation and lack of incidence study and the need for standardized and population based DPN studies in LAC. The statistical analysis methods used are rigorous and appropriate.

Minor points:

1) The Authors could provide more evidence and details on why they included papers regarding studies diabetes type I or both diabetes type II and type I. Diabetes type I and II have different pathophysiology, age of onset etc. Hence it is important to justify their choice or remove these studies (in total 5, 3 type I and 2 both Type1 and 2) from the analysis. Furthermore, the numbers for diabetes type I are too to make any conclusion about the differences in prevalence between the 2 types.

2) The Authors should better justify why they included additional records identified through other sources (Fig 1).

3) Given that neuropathic pain associated with diabetic neuropathy has a great impact on quality of life and health costs, it would be helpful if the Authors could analyze the data separately for painful vs non-painful diabetic neuropathy in this study or in future studies.

4) The Authors should consider to add Fig S1 as one of the main figures of the manuscript and not as supplemental figure. The map gives an immediate view of the countries in within LAC where studies were identified. As stated and discussed by the Authors studies were identified only in 5 countries of the 33 countries in LAC.

6. PLOS authors have the option to publish the peer review history of their article (what does this mean?). If published, this will include your full peer review and any attached files.

Reviewer #1: No

Reviewer #2: **Yes: **Rayaz Ahmed Malik

Reviewer #3: No

---

## [Author Response · Author response to Decision Letter 0]

29 Jan 2021

RESPONSE TO DECISION LETTER:

Paper ID: PONE-D-20-28188R1 

Prevalence and incidence of diabetic peripheral neuropathy in Latin America and the Caribbean: a systematic review and meta-analysis.

Reviewer #1: 

Abstract

1-PubMed is not a database; it is a search engine. You searched MEDLINE via PubMed.

** R. We corrected it.

2-Mention the used program for analysis

** R. We added it.

3-During reporting numbers with fractions or decimals, please use (.) not (,).

** R. We corrected it. 

4-P-value here 48.5% (95%CI: 40.2-56.8; I2=97.94%; p<0.01) for heterogeneity or effect size?

**R. The P-value was for heterogeneity. We corrected that. 

Introduction

5-Well-written with clear rationale and objectives.

**R. Thanks for your feedback. 

Methods

6- "A systematic review with meta-analysis was performed to assess the prevalence of DPN in patients with DM." Please remove this sentence as it was already mentioned in the objectives.

** R. We removed this sentence.

7- In your protocol, you proposed to identify the prevalence and incidence of "painful" DPN in Latin America, while in the manuscript you aimed to identify the prevalence and incidence of DPN in Latin America and the Caribbean. Please mention this deviation in your manuscript or update the protocol.

** R. We updated the Prospero protocol:

https://www.crd.york.ac.uk/prospero/display_record.php?RecordID=148273

The outcome was modified by changing the painful neuropathy to peripheral neuropathy due to the little valid methods and high heterogeneity for its evaluation in the region. The population is better specified by adding the Caribbean region to Latin America. The inclusion criteria of the studies were specified according to the diagnostic methods of peripheral neuropathy. The analyzes to assess heterogeneity were better detailed.

8- The inclusion criteria of the population is not clear.

** R. We improved the redaction of the inclusion criteria section.

9- It's not clear to me why you excluded studies that reported some comorbidities such as diabetic foot?

** R. We excluded subsequent stages of diabetes mellitus such as diabetic foot or hospitalized for to reduce the selection bias of samples with expected high prevalence of DPN.

10- Databases were searched one year ago, I strongly recommend re-searching the databases for any potential studies.

** R. We updated the search on December 14th, 2020 for publication purposes. We found 164 papers and added 4 articles in the analysis. We modified all tables and figures accordingly.

11- Did you plan to include case reports and case series? If not, please specify.

**R. We did not plan to include them. This is described in the protocol and we added an exclusion criterion about it in the manuscript.

12- Two independent authors (MMY y MOF). You may mean (MMY and MOF).

** R. Corrected

13- Line 132: Data >> were not was

**R. Corrected.

14- Please recheck these percentages "quantified using I2 statistical test considering that an I2 < 40% is low, 30-60% is moderate, 50-90% is substantial, and 75-100% is considerable heterogeneity": 

**R. Thank you for your suggestion. We have checked the Cochrane Handbook and it suggests these thresholds to interpretate heterogeneity. We choose threshold for considerable heterogeneity of 75% according to Higgins et al. Although there is not statement about these limits.

Reference: https://handbook-5-1.cochrane.org/chapter_9/9_5_2_identifying_and_measuring_heterogeneity.htm

15- Authors did not discussed who did they handle the duplicates and missing data.

Results

** R. We modified the paragraph in the methods section:

According to the inclusion criteria, two independent authors (MMY y MOF) selected articles by titles and abstracts to identify potentially relevant articles. One of the authors (MMY) handled the duplicates. Lastly, the same authors accessed the full-text articles and evaluated their eligibility for inclusion. A third author (MYA) addressed the missing data and resolved inclusion discrepancies by discussion and consensus.

16- Line 213: There is no supplementary file 8, you may mean S2 Table?!

**R. Corrected

17- Line 248: It is 48.5% not 48,5%

**R. Corrected

18- Line 251: 11.9% not 11,9%. This should be addressed in the entire manuscript

**R. Corrected in all document

19- I could not find S6 Table??!

**R. The numeration of all tables was reviewed and corrected.

20- The quality of the figures should be enhanced

**R. We improved the quality of the figures.

21- Is there any explanation for this specific sample size (Sample size (>323) In the meta regression?

**R. We explained this in Risk Bias assessment section. : “adequate sample size > 323 subjects, considering the prevalence of DPN of 30% according to Sun et al. [27], 5% alpha, and 80% of power”. 

We added this sentence in the table 4 and 5 as footnote.

Discussion

22- Unresolved heterogeneity should be reported in the limitations.

**R. We added in the limitation section the sentence between quotation marks:

The present study has some limitations. We only identified studies from five countries, of a total of 33 countries in LAC. Moreover, most of the data was based on a hospital population, with limited general population participation. Therefore, the external validity of our estimates has to be interpreted with caution. “Furthermore, although we have identified several sources of heterogeneity of the pooled estimates, a large unresolved heterogeneity was found”. 

23- In the conclusion, you have to mention that this is a very low evidence.

** R. We added in the conclusion the words between quotation marks:

This study revealed that the overall prevalence of DPN was relatively high in LAC countries compared to other regions, as almost half of DM patients presented DPN, “although from very low evidence.” 

General comment.

24- Please consider a language editing to eliminate any language mistakes

**R. The manuscript was reviewed by an English native speaker and we corrected all language mistakes.

Reviewer #2: 

This is an important contribution as it is a systematic review and metanalysis of the incidence and prevalence of DPN in Latin America and the Caribbean, a region with an explosion of diabetes and its complications and as highlighted a real lack of quality data on DPN.

The methods employed for the SR and MA are appropriate.

It highlights the high prevalence of this often neglected complication of diabetes and highlights the lack of rigorous population based studies to define the prevalence of DPN in LAC and the Caribbean.

It also reflects the limited published data from the region and the wide ranging prevalence (13-93%) due to differing populations studied, especially those from secondary care (n=19/25) and the different definitions utilized to diagnose DPN. Only one small study that assessed the incidence is noted. It confirms the poor methods used to identify DPN with most studies only being able to identify probable DPN as they rely on symptoms and clinical deficits.

The title should include the word Caribbean.

R. We added the word in the manuscript

What about painful DPN?

** R. Thank you for your feedback. The clinical manifestations of DPN were not prioritized outcomes in our predefined protocol, therefore, in this study, we focus only on the prevalence and incidence of DPN and their factors of heterogeneity. However, our research group is working in an in-depth review on the reported clinical profile of DPN in the region using a more appropriate methodological approach. 

Reviewer #3: 

The manuscript is a systematic Review and meta-analysis to estimate the prevalence and incidence of Diabetic Peripheral Neuropathy (PDN) in Latina America and the Caribbean (LAC). The Authors demonstrated that in LAC there is a high (49.5%) prevalence of PDN, a significant health issue. Furthermore, the Authors identify significant heterogeneity between and within country. The results are convincing and indicated several gaps, including under-representation and lack of incidence study and the need for standardized and population based DPN studies in LAC. The statistical analysis methods used are rigorous and appropriate.

Minor points:

1) The Authors could provide more evidence and details on why they included papers regarding studies diabetes type I or both diabetes type II and type I. Diabetes type I and II have different pathophysiology, age of onset etc. Hence it is important to justify their choice or remove these studies (in total 5, 3 type I and 2 both Type1 and 2) from the analysis. Furthermore, the numbers for diabetes type I are too to make any conclusion about the differences in prevalence between the 2 types.

**R. Thank you for your comments. We have clarified the justification for inclusion of T1DM and mixed population in our analysis. Please see the inclusion criteria section.

In the results section, as you suggested, we added, in the subgroup analysis, a statement about T1DM (between quotation marks):

The subgroup analysis for the type of DM showed no differences (heterogeneity test between-groups, p=0.53) among patients with type 1 DM (54.8%, 95% CI: 30.8 to 77.7) compared to type 2 DM (44.8%, 95% CI: 35.4 to 54.4), “however, this comparison is underpowered due to small number of studies including type 1 DM patients (n=3).”

In the discussion section, we already mentioned the explanation about T2DM an T1DM differences:

“Regarding the DM type, due to the small number of studies on type 1 DM and its wide confidence interval, it is not possible to establish differences in prevalence between both types. Nonetheless, in a recent meta-analysis by Sun et al., they found that patients with type 2 DM presented a higher DPN prevalence than those with type 1 DM [27]. This disparity may be explained by the differences in the pathophysiology of DPN between both types. In type 2 DM, dyslipidemia, insulin resistance, and systemic inflammation are significant factors for DPN, which can develop before diabetes onset and diagnosis [16]. In contrast, type 1 DM onset is mostly correlated with the presentation of symptoms, caused primarily by insulin deficiency and hyperglycemia, which is why a tight glucose control could reduce the risk of DPN in type 1 DM, but not in type 2 DM [81].”

Finally, we also added this paragraph in the discussion section about mixed diabetes studies:

“Since we decided to included studies with mixed populations (T1DM or/and T2DM) in our overall pooled estimates in order to increase generalizability and to reduce the impact of potential diagnosis overlap between T1DM and T2DM, we performed a sensitivity analysis to explore heterogeneity, that showed non-important impact of T1DM population into the regional pooled estimates (likely owing to the small number of included studies and sample size), contrary to previous studies showing less prevalence of DPN in T1DM (1). However, it is important to considering that current studies postulate that the diabetes mellitus classification is insufficient, and it has been suggested 5 phenotypes that would better explain the long-term results (2). Moreover, the Latent Autoimmune Diabetes in Adults (LADA) could simulates T2DM onset and turns to a total insulinopenia in a short-term period (3), thus complicating the DM type differentiation. Additionally, the unavailability of laboratory tests for T1DM-specific antibodies in LAC countries could hamper the correct diagnosis (4). Therefore, it is important to increase the number and quality of the T1 DM diabetes registries in LAC, to estimate a precise DPN prevalence in this population.” 

References: 

1. Hicks CW, Selvin E. Epidemiology of Peripheral Neuropathy and Lower Extremity Disease in Diabetes. Curr Diab Rep. 2019;19(10):86. Published 2019 Aug 27. doi:10.1007/s11892-019-1212-8

2. Ahlqvist E, Storm P, Käräjämäki A, Martinell M, Dorkhan M, Carlsson A, et al. Novel subgroups of adult-onset diabetes and their association with outcomes: a data-driven cluster analysis of six variables. The lancet Diabetes & endocrinology. 2018;6(5):361-9.

3. American Diabetes A. 2. Classification and Diagnosis of Diabetes: Standards of Medical Care in Diabetes—2021. Diabetes Care. 2021;44(Supplement 1):S15-S33.

4. Vento S, Cainelli F. Autommune Diseases in Low- and Middle-Income Countries: A Neglected Issue in Global Health. Isr Med Assoc J. 2016 Jan;18(1):54-5. PMID: 26964282.

2) The Authors should better justify why they included additional records identified through other sources (Fig 1).

We changed in the search section:

Instead of this: “We conducted a hand search of grey literature and other related articles to retrieve additional relevant articles.”

Change for this: “As recommended by Cochrane collaboration, we included additional relevant articles from other sources via a hand search of grey literature and other related articles, due to low rate of database indexation of regional journal (1).” 

Reference: 

1. Rodrigues RS, Abadal E. Ibero‐American journals in Scopus and Web of Science. Learned publishing. 2014;27(1):56-62.

3) Given that neuropathic pain associated with diabetic neuropathy has a great impact on quality of life and health costs, it would be helpful if the Authors could analyze the data separately for painful vs non-painful diabetic neuropathy in this study or in future studies.

**R. Thank you for your feedback. The clinical manifestation of DPN was not prioritized outcomes in our predefined protocol, therefore, in this study, we focus only on the prevalence and incidence of DPN and their factors of heterogeneity. However, our research group is working in an in-depth review on the reported clinical profile of DPN in the region using a more appropriate methodological approach. 

4) The Authors should consider adding Fig S1 as one of the main figures of the manuscript and not as supplemental figure. The map gives an immediate view of the countries in within LAC where studies were identified. As stated and discussed by the Authors studies were identified only in 5 countries of the 33 countries in LAC.

**R. Thanks for your suggestion. We have added the map as a main figure in manuscript.

---

## [Decision Letter · Decision Letter 1]

30 Mar 2021

PONE-D-20-28188R1

Prevalence and incidence of diabetic peripheral neuropathy in Latin America and the Caribbean: a systematic review and meta-analysis.

PLOS ONE

Dear Dr. Yovera-Aldana,

Thank you for submitting your manuscript to PLOS ONE. After careful consideration, we feel that it has merit but does not fully meet PLOS ONE’s publication criteria as it currently stands. Therefore, we invite you to submit a revised version of the manuscript that addresses the points raised during the review process.

We look forward to receiving your revised manuscript.

Kind regards,

Ahmed Negida, MD

Academic Editor

PLOS ONE

Journal Requirements:

Reviewers' comments:

Reviewer's Responses to Questions

**Comments to the Author**

1. If the authors have adequately addressed your comments raised in a previous round of review and you feel that this manuscript is now acceptable for publication, you may indicate that here to bypass the “Comments to the Author” section, enter your conflict of interest statement in the “Confidential to Editor” section, and submit your "Accept" recommendation.

Reviewer #1: All comments have been addressed

Reviewer #2: All comments have been addressed

Reviewer #3: All comments have been addressed

2. Is the manuscript technically sound, and do the data support the conclusions?

Reviewer #1: Yes

Reviewer #2: Yes

Reviewer #3: Yes

3. Has the statistical analysis been performed appropriately and rigorously? 

Reviewer #1: Yes

Reviewer #2: Yes

Reviewer #3: Yes

4. Have the authors made all data underlying the findings in their manuscript fully available?

Reviewer #1: Yes

Reviewer #2: Yes

Reviewer #3: Yes

5. Is the manuscript presented in an intelligible fashion and written in standard English?

Reviewer #1: Yes

Reviewer #2: Yes

Reviewer #3: Yes

6. Review Comments to the Author

Reviewer #1: The manuscript has been improved significantly; however, I have two minor comments that have not been addressed yet.

1- Line 121: add and between (MMY MOF)

2- Line 116: Not for "Publication purposes", it is to find any potentially eligible studies to be included.

3- Line 370: Correct this presentation to avoid any confusion: 46.5%(95%CI; 38.0 to 55.0, I2=98.2%; p<0.01)

4- Line 371: I2: Not calculated? Why? It can be calculated if you have two or more studies compared, to the best of my knowledge.

5- Line 381: 35.78% (95% CI: 27.86 to 44.55; I2), add the I2 and its p-value in a separate ()

6- Line 381: (p=0.000), correct this to be (p<0.001)

Reviewer #2: My comments have been addressed adequately.

This is an important contribution, not least because it highlights the lack of good prevalence studies from the region.

Reviewer #3: The Authors addressed adequately all the Reviewer's comments. All concerns were addressed in the edited version of manuscript

7. PLOS authors have the option to publish the peer review history of their article (what does this mean?). If published, this will include your full peer review and any attached files.

Reviewer #1: No

Reviewer #2: **Yes: **Rayaz Ahmed Malik

Reviewer #3: No

---

## [Author Response · Author response to Decision Letter 1]

7 Apr 2021

Response to reviewers

PONE-D-20-28188R2

Prevalence and incidence of diabetic peripheral neuropathy in Latin America and the Caribbean: a systematic review and meta-analysis.

PLOS ONE

Reviewer #1: The manuscript has been improved significantly; however, I have two minor comments that have not been addressed yet.

1- Line 121: add and between (MMY MOF)

***R: Corrected

2- Line 116: Not for "Publication purposes", it is to find any potentially eligible studies to be included.

***R: Corrected.

3- Line 370: Correct this presentation to avoid any confusion: 46.5%(95%CI; 38.0 to 55.0, I2=98.2%; p<0.01)

*** R: Corrected : 46.5% (95%CI: 38.0 to 55.0; I2=98.2%, p<0.01)

4- Line 371: I2: Not calculated? Why? It can be calculated if you have two or more studies compared, to the best of my knowledge.

*** R: Thank for your comment. We agree with you that it is mathematically possible to calculate the I2 in meta-analysis of 2 or more included studies. However, we used the metaprop command in Stata (1), and the package developers only provide the I2 for meta-analysis of 4 or more studies. This is justified by previous studies suggesting a potential high bias of the I2 statistic in small meta-analyses (2), and due to this uncertainty, it is more accurate to assess the heterogeneity assessing visually the confidence interval overlaps (3). Therefore, we decided not to calculate the I2 in the analysis with less than 4 included studies. We have clarified this point in the method section as you suggested. 

References:

1. Nyaga VN, Arbyn M, Aerts M. Metaprop: a Stata command to perform meta-analysis of binomial data. Arch Public Health. 2014 Nov 10;72(1):39. doi: 10.1186/2049-3258-72-39. PMID: 25810908; PMCID: PMC4373114.

2. von Hippel PT. The heterogeneity statistic I(2) can be biased in small meta-analyses. BMC Med Res Methodol. 2015;15:35. Published 2015 Apr 14. doi:10.1186/s12874-015-0024-z

3. Ioannidis JPA, Patsopoulos NA, Evangelou E. Uncertainty in heterogeneity estimates in meta-analyses. BMJ. 2007;335(7626):914–6. doi: 10.1136/bmj.39343.408449.80. 

5- Line 381: 35.78% (95% CI: 27.86 to 44.55; I2), add the I2 and its p-value in a separate ()

*** R: We completed the data: 35.78% (95% CI: 27.86 to 44.55; I2 =99%, p<0.001 )

6- Line 381: (p=0.000), correct this to be (p<0.001)

*** R: Corrected : 30% (95% CI: 25 to 34; I2 = 99.5%, p<0.001)

Reviewer #2: My comments have been addressed adequately.

This is an important contribution, not least because it highlights the lack of good prevalence studies from the region.

Reviewer #3: The Authors addressed adequately all the Reviewer's comments. All concerns were addressed in the edited version of manuscript

---

## [Editor Report · Decision Letter 2]

30 Apr 2021

Prevalence and incidence of diabetic peripheral neuropathy in Latin America and the Caribbean: a systematic review and meta-analysis.

PONE-D-20-28188R2

Dear Dr. Yovera-Aldana,

We’re pleased to inform you that your manuscript has been judged scientifically suitable for publication and will be formally accepted for publication once it meets all outstanding technical requirements.

Kind regards,

Ahmed Negida, MD

Academic Editor

PLOS ONE
---

## [Editor Report · Acceptance letter]

5 May 2021

PONE-D-20-28188R2 

Prevalence and incidence of diabetic peripheral neuropathy in Latin America and the Caribbean: a systematic review and meta-analysis. 

Dear Dr. Yovera-Aldana:

I'm pleased to inform you that your manuscript has been deemed suitable for publication in PLOS ONE. Congratulations! Your manuscript is now with our production department. 

Kind regards, 

on behalf of

Dr. Ahmed Negida 

Academic Editor

PLOS ONE